# A biodegradable, flexible photonic patch for in vivo phototherapy

Kaicheng Deng[1,16], Yao Tang[2,3,16], Yan Xiao [4,16], Danni Zhong[5,16], Hua Zhang[6], Wen Fang[1], Liyin Shen[1], Zhaochuang Wang[4], Jiazhen Pan[6], Yuwen Lu[1], Changming Chen[7], Yun Gao[6], Qiao Jin[6], Lenan Zhuang[6], Hao Wan[7], Liujing Zhuang[7], Ping Wang[7], Junfeng Zhai[8], Tanchen Ren[9], Qiaoling Hu[1], Meidong Lang[4], Yue Zhang[10], Huanan Wang[6], Min Zhou [3,5,11,12 ✉], Changyou Gao [1,13 ✉], Lei Zhang [2,3 ✉] & Yang Zhu [1,14,15 ✉]

Diagnostic and therapeutic illumination on internal organs and tissues with high controllability and adaptability in terms of spectrum, area, depth, and intensity remains a major challenge. Here, we present a flexible, biodegradable photonic device called iCarP with a micrometer scale air gap between a refractive polyester patch and the embedded removable tapered optical fiber. ICarP combines the advantages of light diffraction by the tapered optical fiber, dual refractions in the air gap, and reflection inside the patch to obtain a bulb-like illumination, guiding light towards target tissue. We show that iCarP achieves large area, high intensity, wide spectrum, continuous or pulsatile, deeply penetrating illumination without puncturing the target tissues and demonstrate that it supports phototherapies with different photosensitizers. We find that the photonic device is compatible with thoracoscopy-based minimally invasive implantation onto beating hearts. These initial results show that iCarP could be a safe, precise and widely applicable device suitable for internal organs and tissue illumination and associated diagnosis and therapy.

Illuminating internal organs and tissues has widely supported optical diagnosis, laser surgery, and other light-activated therapies by creating unique lightened microenvironments on targets[1-7]. Improving the precision, versatility, and controllability in delivering photo energy is the key to increasing the safety, efficacy, and scope of application of light-based procedures[8,9]. So far, available in vivo illumination strategies have not universally met the depth, area, wavelength, and power requirements across different working scenarios, particularly in achieving large-area illumination on deep targets[10-12]. Light from external sources can hardly reach deep organs and tissue due to absorbance and scattering of photons, and autofluorescence, even for light in near-infrared I and II windows[13,14]. Therefore, placing light sources on target in vivo is the mainstream solution for illumination of the heart, brain, liver, etc[15-19]. Recent studies have employed implantable near-field-communication (NFC)-based light-emitting-diode to wirelessly transfer ex vivo magnetic power into optical energy[20], or upconversion nanoparticles[21-23] to transfer long-wavelength light to short-wavelength light in situ, for photodynamic therapy and fluorescent imaging applications. In these promising designs, illumination intensity limits are determined by the energy-transferring efficiency of the NFC device, or the light penetration and quantum-transferring efficiencies, respectively.

Optical fiber devices support precise, wide-spectrum, high-power waveguiding, and can be fabricated from clinically approved metal-free biocompatible materials[24-26], hence are attractive photonic devices for illuminating internal organs and tissues. Integrated with endoscopes, or guided by magnetic resonance imaging, optical fiber devices could be minimally invasively placed at desired locations, for targeted illumination in applications including laser-induced thermotherapy, optogenetics, temperature and pressure sensing, etc[27-29]. For deep targets, optical fibers need to be inserted into the tissues, which is associated with risks of impairing tissue functions[30-32]. Through the

structural design of the optical fiber-based photonic devices, optical energy distribution can be manipulated to some extent[25,33–35]. The tips of regular optical fibers are flat, therefore light leaving the fiber tips transmits in forward direction, only illuminates a small piece of tissue in front of the fiber tip[30,34]. Surface etching and side polishing could increase emission on radial directions of the optical fibers, but only around the modified region, thus covering a column-like volume, in which the radius is the light penetration depth, and the height is the length of the modified region plus the light penetration depth[30,36]. In optogenetics applications, the optical fibers are often tapered to precisely concentrate light in a defined region[30]. Attaching a planar poly(L-lactic acid) waveguide at the optical fiber tip which puncture into target tissue instead of the optical fiber could spread optical energy in the front direction, corresponding to the shape of the cross-section[37–39]. Despite the design and modification strategies, optical fiber-based large area, deep, non-destructive illumination with controllable power and wavelength is still highly challenging, particularly on constantly moving targets like heart, lungs, and muscles[15,40,41].

In this work, we present a biodegradable, flexible, photonic device, iCarP, for internal organ and tissue illumination (Fig. 1). In iCarP design, a flexible tapered optical fiber (TOF) was embedded in a biodegradable, transparent polyester (PMCL) patch, and a micrometer-thin air gap was created between TOF and PMCL (Fig. 1a). In this integrated optical device design, the diffraction in the TOF end, refractions on the air/TOF and air/PMCL interfaces collectively enhance the scattering of delivered optical energy (Fig. 1a). Unlike most optical fiber-based photonic devices, iCarP with the embedded TOF goes parallelly to the surface of the target tissue, instead of being inserted into the tissue. As a result, large area, deeply penetrating illumination without invasively damaging target tissues has been achieved (Fig. 1b). Light scattering efficiency of iCarP is first evaluated in vitro and in silico. The controllability and robustness of iCarP illumination, in terms of supported spectrum, intensity, area, and temporal pattern are demonstrated in mice tumor photothermal and

phototherapy treatments, and a rat myocardial infarction (MI) photosynthesis treatment model. Compatibility with photosensitizers of different functions and absorption wavelengths is tested. Improvement in therapeutic outcome by the light scattering effect of iCarP is evaluated compared to clinically relevant optical fiber structures. The stability of iCarP illumination on mechanically challenging, cyclically contracting myocardium is studied in situ photosynthesis for MI treatment (Fig. 1c, d). TOF removal after repeated illuminations is also experimented in vivo. Compatibility of iCarP with minimally invasive implantation and illumination is tested in a canine thoracoscopy surgery[42].

## Results

### Light scattering by iCarP

The light scattering capacity marked by the light divergence angle (the angle between the two edges of the output beam) of photonic medical devices is a key factor in determination of the scope of tissue illumination. Light emitting from optical fibers with flat-end fiber ends mainly converged in the forward direction, generating a flashlight-like illumination mode, and a corresponding small illumination area. In addition, the flashlight-like illumination mode usually requires the optical fiber to pierce the target tissue. Therefore, it is desired to turn the flashlight-like illumination mode into a bulb-like illumination mode featured with large divergence angles, so that increased lateral illumination from optical fiber-based photonic devices parallelly implanted onto organ/tissue surfaces could support large area, deep, and noninvasive illumination. In iCarP design, all three components (TOF, flexible patch, and particularly, the air gap) serve to enhance light scattering performance.

In order to obtain larger divergence angles, optical fibers were tapered first to generate a fiber tip geometry more suitable for light scattering (Fig. 2a) compared to flat-end optical fibers (Fig. 2b). Unlike the commercial or tapered optical fiber reported in previous studies[30,41], our TOF has a much shorter taper length (280 μm) to

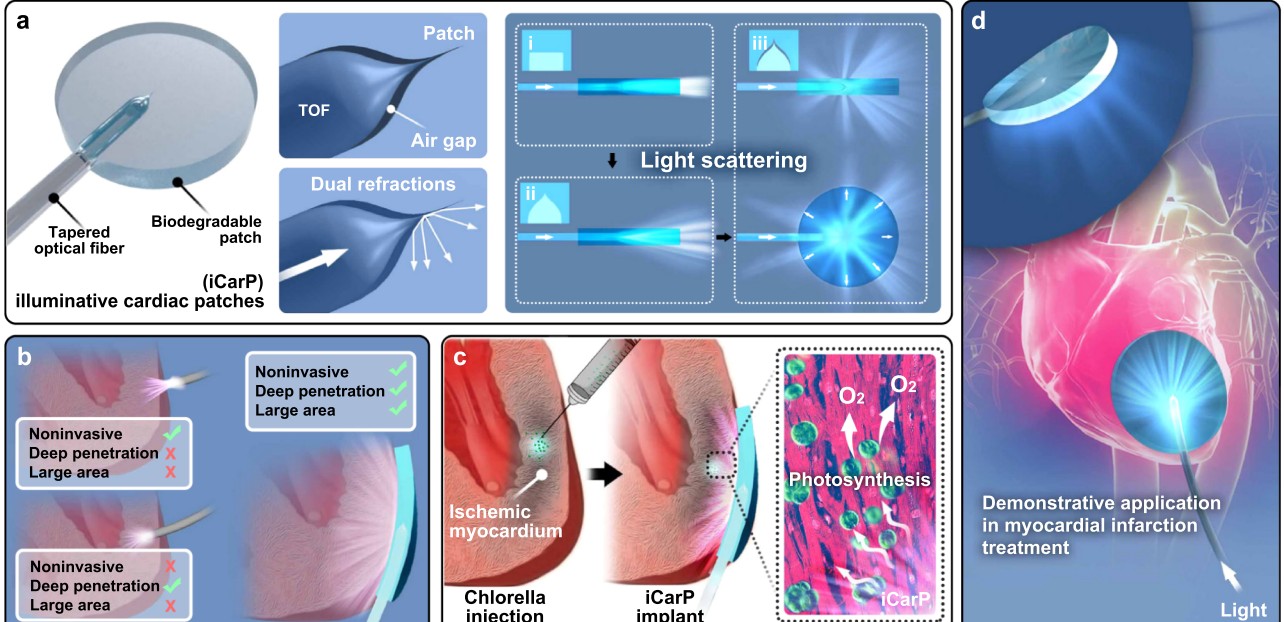

**Fig. 1 | The design, function and demonstrative application of the light scattering photonic device, illuminative cardiac patch (iCarP) for large area, deep illumination on internal organs and tissues. a** Structure and key components of iCarP. An air gap was created between the tapered optical fiber (TOF) and the waveguiding patch substrate. The diffraction by TOF, dual refractions on TOF/air and air/patch interfaces of the air gap contribute to significantly greater light

scattering, particularly, lateral illumination compared to configurations of flat-end optical fiber without air gap, and TOF without air gap. **b** Comparison between illumination effects by optical fiber only (on top or inside myocardium) and iCarP. ICarP supports noninvasive, deep, and large-area illumination. **c, d** Illumination on intramyocardially injected chlorella by iCarP induces in situ photosynthesis in infarcted myocardium and its application in myocardial infarction treatment.

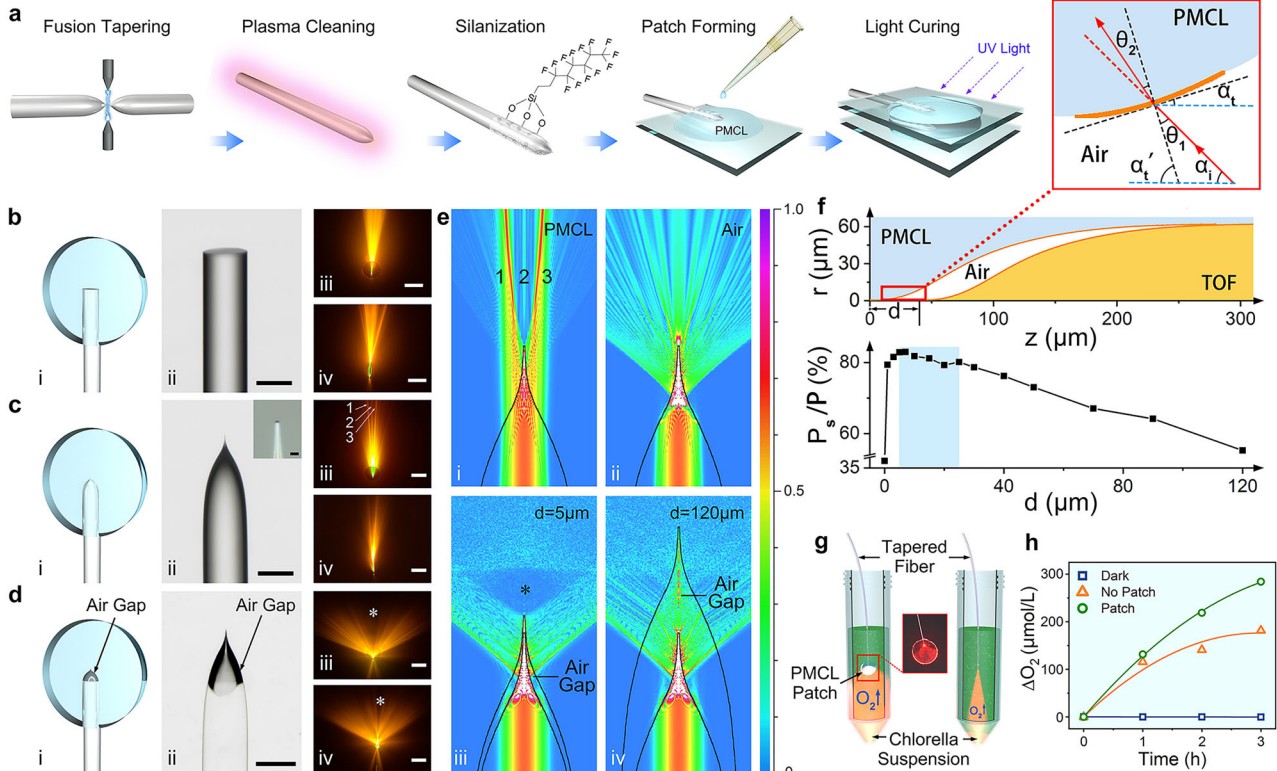

**Fig. 2 | Fabrication and light scattering performance of iCarP. a** Fabrication of TOF in PMCL patch. An air gap was generated to obtain iCarP. **b–d** Photonic devices of flat-end optical fiber (FEOF) (**b**), tapered optical fiber (TOF) embedded in PMCL patch (**c**), and iCarP (**d**). (i) schematic diagram of the devices, (ii) microscopic view of the devices (representative for > 3 devices), scale bar = 100 μm, inset of **c**: tapered fiber tip, scale bar = 2 μm, (iii, iv) front view (iii) and side view (iv) of the photonic devices in (i) and fluorescence excited by emitted light, scale bar = 5 mm. * indicates the dark region in forward direction of iCarP. scale bar = 5 mm. **e** Simulated electric field intensity distribution of light emitted from the photonic devices, (i) TOF in PMCL without air gap, (ii) TOF in air, (iii) iCarP with d of 5 μm, (iv) iCarP with d of 120 μm. **f** Top: the axisymmetric cross-section diagram of the air gap and the tip of TOF in iCarP, bottom: the relationship between the percentage of lateral optical power ($P_s$) to total power (P) and d, inset: local light path diagram, (z: distance between the TOF tip to the tip of air gap; $\alpha_t$: the angle between the tangent plane of the air/PMCL interface and the axial direction of the TOF; $\alpha_t'$: the angle between the normal line of the tangent plane and the axial direction of the TOF; $\alpha_i$: the divergence angle of incident light; $\theta_1$: angle of incidence on air side; $\theta_2$: angle of refraction on PMCL side; P: total optical power out of air gap; $P_s$: optical power out of forward propagation region). **g** Scheme of detection of oxygen production of chlorella induced by iCarP illumination: iCarP (left) and TOF (right) at "on" state were immersed in the chlorella suspension (green), and the oxygen production was monitored. **h** Cumulative oxygen production of chlorella induced by iCarP or TOF illumination, and in dark.

increase light scattering, the tip is 1 μm in diameter, as shown in Fig. 2c (ii). The shape of the TOF tip can be described by Supplementary Eqn. 1. As the fiber core narrows down, the cross-sectional diameter of the TOF decreases rapidly at the tip. When light propagates forward along the TOF tip, more light is forced to emit from the tip side, because the tip size is down to 1 μm, the diffraction was significant in the air that the light emitted at a larger divergence angle. The measured divergence angles of TOF in PMCL and air were 40° (Fig. 2c). The divergence angles of TOF in the air given by the simulation model was 97° (Fig. 2e, ii). Both angles were greater compared to the divergence angles of flat-end optical fiber in PMCL (20°, Fig. 2biii, iv).

In iCarP, TOF was parallelly embedded in the PMCL substrate, an air gap was created on the interface of the TOF tip and the PMCL substrate by carefully pulling the TOF out of the PMCL substrate under the microscope (Fig. 2d, i, ii). Since the external (material surrounding the TOF) refractive index is crucial to the output characteristics of optical device, the air gap design plays an important role in increasing the divergence angle. Simulation result in Fig. 2e (i) revealed that without the air gap, light entering the PMCL substrate from the TOF mainly concentrates in three forward going beams, which was confirmed in the light path experiment, as indicated by numbers 1, 2, 3 in Fig. 2c (iii). In addition to the significantly greater divergence angle on TOF/air interface (97°, simulated) compared to that on TOF/PMCL interface (40°), TOF was predicted to have a more uniform distribution

of light in air (no high-intensity regions in Fig. 2d as indicated by red or yellow color, compared to the distribution pattern of TOF in PMCL shown in Fig. 2c). Therefore, we built an air gap around the tip of the TOF in order to obtain a large divergence angle on TOF/air interface and large incidence angle on air/PMCL interface, since it is the premise of achieving high-intensity lateral illumination from iCarP.

As the light emitted from the TOF reaches the PMCL substrate, secondary refraction would occur on the air/PMCL interface, which is determined by the refractive indexes of air and PMCL ($n_{air} \times \sin\theta_1 = n_{PMCL} \times \sin\theta_2$, $\theta_1$ and $\theta_2$ were the angle of incidence light into PMCL and angle of refraction on PMCL side, respectively). Due to the side methyl molecular structure, PMCL, the crystallinity was significantly reduced compared to semi-crystalline PCL, which resulted in high transparency of the PMCL patch substrate. In addition, the double bond-based covalent cross-linking fixed the amorphous status. When a light incident on the air/PMCL interface, the high refractive index of PMCL ($n_{PMCL} = 1.49$) resulted in a large angle refraction, which further increased the divergence angle (130°), as measured in light path experiments (Fig. 2d) and demonstrated in simulation results (Fig. 2e, iii, iv). As the result of refraction on air/PMCL interface, not only the divergence angle increased, achieving the desired wider bulb-like illumination, the optical energy distribution was more even compared to TOF in air. The simulation indicates that optical energy in the forward direction of iCarP is lower than average within the illuminated

space, forming a dark region (Fig. 2e, iii, iv, marked by *). Consistently, experimental results confirmed the existence of the dark region (*) in the forward direction, in both top-down view and lateral view of iCarP (Fig. 2d, iii, iv). These findings confirmed that the delivered light was scattered into a bulb-like illumination mode by consecutive refractions on the TOF/air interface and air/PMCL interface, and guided away from the forward direction, to the direction perpendicular to the patch plane of iCarP device, which is desired as the tissue covered by the implanted iCarP will be in the perpendicular direction of the device.

In addition to the optical properties of the materials on both sides of the air gap, the geometry of the air gap, dictated by the distance between the TOF tip and air gap tip ($d$), is a critical factor determining the light scattering performance of the air gap featured with the two light-refracting interfaces. We simulated the influence of $d$ (0 - 120 μm) on the performance of light refraction and distribution. From 43° (the critical angle of air/PMCL interface) clockwise to 43° counterclockwise of the optical fiber axis is defined as the forward propagation region, while the tip of TOF is the vertex. The ratios between $P_s$ (optical power outside the forward propagation region) and P (total optical power out of air gap) at different $d$ were given by the computational model. As shown in Supplementary Fig. 3, in the range of 5–25 μm, the iCarP created a similar distribution pattern of emitting light, showing greater $P_s$/P (80%) compared to $d$ < 5 μm and $d$ > 25 μm (Fig. 2f). $P_s$/P decreases as $d$ decreases in the range of 0–5 μm, and as $d$ increases in the range of 25–120 μm. Therefore, 5–25 μm is the target range of $d$ in TOF pull-out process. An explanation of the geometric optics can be given as the following.

As shown in Fig. 2f, air gap structure including the TOF, air, and PMCL was reconstructed in a MATLAB model to derive the dependence of $\alpha_t$ (the angle between the tangent plane of the air/PMCL interface and the axial direction of the TOF) on $z$ (distance between the TOF tip to the tip of air gap). In this model, $r$ was the radius of air gap cross-section, $\alpha_{t'}$ was the angle between the normal line of the tangent plane and the axial direction of the TOF, $\alpha_i$ was the 1/2 divergence angle of incident light. As long as $\alpha_i < \alpha_{t'}$, $\theta_2$ will be smaller than $\theta_1$, and light will be further scattered. When $d$ is between 5 μm and 40 μm (covering the 5–25 μm target range), $\alpha_i$ is greater than $\alpha_{t'}$, the light incidence into PMCL could be further refracted, approaching to the radial direction of TOF (perpendicular to the patch), which is desired for illuminating device covered tissue. When $d$ is too big (>40 μm), light with divergence angle of 130° (65° between the light and TOF axial direction) would be incident on air/PMCL interface with $\alpha_{t'}$ smaller than $\alpha_i$. As $d$ increases, despite that the divergence angle of TOF in air gap remains at 130°, bigger portion of light (from 130° to smaller divergence angles) would have $\alpha_i < \alpha_{t'}$, less light would be further scattered on air/PMCL interface. On the other hand, if the $d$ is too small (<5 μm), more light with small divergence angle would incident on the flat tip of the air gap (PMCL side) and would not be refracted to a larger divergence angle.

As an in vitro demonstration of the functional value of light scattering, Chlorella suspension were illuminated by iCarP or tapered optical fibers to verify the increase in photosynthesis rate by light scattering achieved via combining TOF tip design, air gap, and PMCL substrate. Chlorella suspension did not produce oxygen in a completely dark environment (Fig. 2g, h). TOF immersed in Chlorella suspension illuminated a narrow cone region in the suspension, as shown in Fig. 2g. In the first hour, Chlorella produced about 115.6 μmol/L oxygen, but the photosynthesis rate significantly decreased in the following 2 h, only about 66.0 μmol/L oxygen were generated. In contrast, iCarP immersed at the same depth illuminated a significantly greater portion of Chlorella in the suspension (Fig. 2g). At the same illumination power, the scattered light increased by 10% oxygen production in the first hour compared to TOF only group (Fig. 2h). A greater improvement photosynthesis rate was observed in the second and third hour of illumination: oxygen produced in the latter 2 h were

more than the first hour in iCarP group, opposite to the observation in TOF only group. A total of 284.2 μmol/L oxygen was generated in iCarP group (Fig. 2h).

## Demonstration of different iCarP illumination modes in vivo

The versatility and controllability of iCarP illumination were evaluated in a rat in vivo model. Using a fast crosslinking biodegradable adhesive (Supplementary Fig. 2), iCarP was firmly adhered to the epicardium, and no detachment was observed after more than 100,000 heartbeats. No interruption in illumination or significant changes in illumination area was observed, indicating the structural integrity of iCarP, including the air gap and TOF tip, was maintained in the rigorous environment: >300 bpm heart rate, >10% strain in each diastole-systole cycle. Therefore, iCarP attachment to other internal organs shall also be firm and able to resist the interference of tissue movements, as shown in a rat model of thigh muscle and liver, (Supplementary Fig. 4).

The flexible optical fiber employed in this study supports the transmission of visible light and infrared light. Consecutive transmission of laser with wavelengths of 405 nm, 520 nm, and 660 nm in the same animal was demonstrated, as shown in Fig. 3a, b. and Supplementary Movie 1. The divergence angles of shorter wavelength lasers were smaller than 660 nm laser, which is believed due to the smaller refractive indexes of iCarP components including the PMCL patch corresponding to the blue and green laser, compared to the refractive index of the red laser. In addition, the compatibility of iCarP with lasers of 445 nm (Supplementary Fig. 5), 473 nm (Supplementary Fig. 6), and 808 nm (Fig. 4) wavelengths was also demonstrated. The wavelengths used above are highly representative of current phototherapies, which showed that iCarP illumination meets the wavelength requirements in photo-responsive drug delivery, photodynamic therapy, photothermal therapy, optogenetics, etc.

Pulsatile illumination at 1 Hz and precise intensity control was achieved by adjusting the pump current of laser diode drivers, combined with wavelength manipulation, showing the programmability and maneuverability of iCarP illumination to meet personalized needs or respond to sudden events (Fig. 3c, d and Supplementary Movie 2). With the enhanced light scattering capability, iCarP was able to illuminate the whole rat left ventricle, particularly at high laser intensity (Fig. 4e, f and Supplementary Movie 3). High intensity laser managed to penetrate the closed chest, which is significantly thicker compared to LV myocardium (Fig. 5b), indicating a full-thickness illumination on LV.

After closing the chest, iCarP was able to continuously transmit light for 3 h (limited by the duration of anesthesia). In one rat, the flexible optical fiber was detached from the laser diode and indwelled in the animal, re-attach to the laser diode 3 h from detachment for the second round of illumination. This showed the feasibility of repeated illumination, and the ability of iCarP to remain functional after being challenged in a closed-chest environment.

## ICarP illumination evaluation ex vivo

In addition to fiber tip geometry and air gap, the PMCL patch also improved light scattering performance. When the PMCL patch is applied to the surface of internal organs and tissues, the large refractive index difference between air ($n_{air}$ = 1.0003) and PMCL ($n_{PMCL}$ = 1.49, higher than $n_{air}$) results in a smaller critical angle (43°) on PMCL/air interface (upper surface of iCarP in Fig. 4a) compared to the critical angle (70°) on PMCL/tissue interface (lower surface of iCarP in Fig. 4a) as the refractive index of tissue ($n_{tissue} \approx 1.4$) was closer to PMCL. As a result, the tendency of light reflection on PMCL/air interface was theoretically greater than that on PMCL/tissue interface. Therefore, the possibility of light exiting the PMCL/tissue surface shall be greater compared to the possibility of light escaping on the PMCL/air interface, and more optical energy will distribute on the tissue side of iCarP.

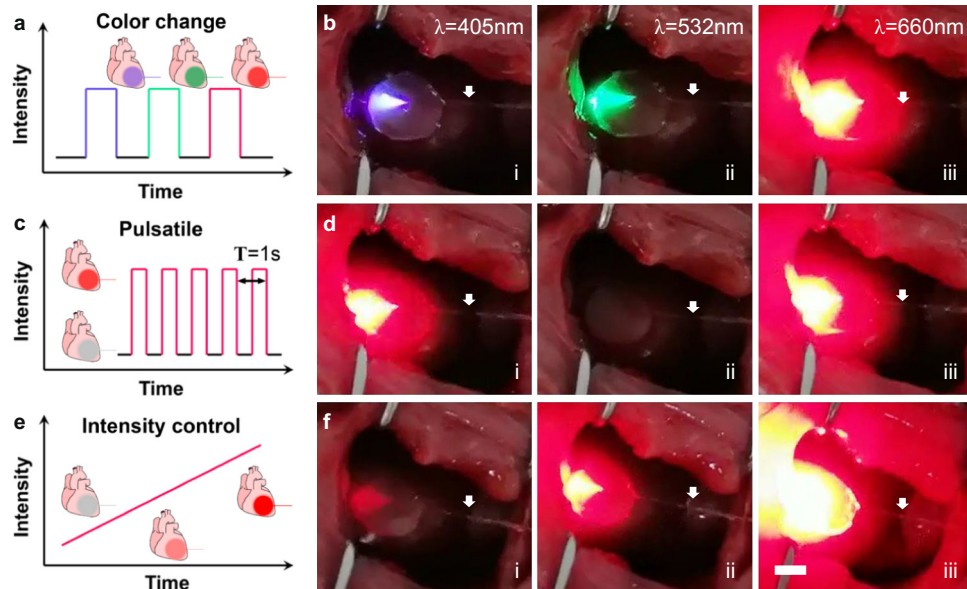

**Fig. 3 | Demonstration of different modes of iCarP illumination on rat hearts.**
**a**, **b** ICarP supporting color change in illumination, the same device delivered light of (i) 405 nm, (ii) 520 nm, (iii) 660 nm. **c**, **d** Pulsatile illumination at 1 Hz, (i), (ii), (iii) shows the on-off-on switch. **e**, **f** ICarP illumination with increasing intensity over time. White arrows indicate the optical fibers.

Device attachment onto excised porcine hearts and their illumination performance were first evaluated. Transmitted laser in PMCL patch devices with flat-end optical fiber ends or tapered optical fibers mainly concentrated in the forward direction (Fig. 4b, c). As a result, only a cone region of the heart was illuminated, on both sides and back of the fiber tips a connected dark region can be found under the PMCL substrate on both photonic devices (Fig. 4b, c). Top view of iCarP illuminated heart showed that the entire iCarP covered area including the area behind the TOF tip, and a 0.5 cm wide outskirt received optical energy (Fig. 4d). The brightest spot located on the TOF tip. A small, relative dark spot was found at the intersection of the forward going direction of the TOF and the edge of the PMCL substrate (Fig. 4d, marked by *), which is consistent with the in vitro result shown in Fig. 2d. Benefit from the high refractive index of PMCL, the illumination scope of iCarP could be adjusted by changing the size of PMCL patch, to match infarcts of different sizes. Attributed to the flexibility of PMCL, iCarP with 2.5 cm diameter could tightly attach to the epicardial surface of porcine hearts, free of detachment in large deformations. These results demonstrated that greater light scattering contributed to the larger illumination area.

Depths of lateral illumination of the devices with flat-end optical fiber ends or TOF, and iCarP were compared in excised porcine myocardium. Red laser (660 nm) from all three devices could penetrate 0.5 cm tissue, but only in iCarP group was laser observed on the opposite side of 1.5 cm thick myocardium, while lateral illumination of devices with flat-end optical fiber ends or tapered optical fibers was not strong enough to penetrate (Fig. 4b–d). In the front view facing the optical fibers, devices embedded optical fibers with flat-end or TOFs created bright elliptical area with >1.5 cm major axis length and > 1 cm minor axis length on the tissue cross-section (Fig. 4b, c). In contrast, light delivered by iCarP emitted from a horizontally wider and vertically lower window on tissue cross section (Fig. 4d), which also evidentially showed that iCarP scattered optical energy in the forward direction of optical fiber to lateral directions, and it works in the tissue environment. These results are consistent with the light distribution profiles of the flat-end optical fibers or TOF measured from −90° to 90° (Supplementary Fig. 7).

The illumination on iCarP outskirt became discrete after lowering the optical power (Fig. 4e). Additional bright spots appeared on the outskirt, which do not fully match with the light paths revealed in Fig. 2d, indicating that internal reflection on the vertical edge inside the PMCL may also support illumination on the entire patch covered region. Laser emission, illumination scope, and depth were not significantly impaired by myocardium distortion, as shown in Fig. 4e(ii). The cross-section closed to the optical fiber of the curved iCarP on the porcine myocardium exhibited the internal optical energy distribution. The penetration depth was uniform across the PMCL patch of iCarP, Fig. 4e(iii). The light emitted from the iCarP illuminated myocardium extended 0.5 cm from the front and rear edges of the PMCL patch, resulted in a circular illuminated area 3 cm in diameter. Laser distribution in porcine myocardium by clinically used matt flat-end optical fibers and side glow optical fibers for tumor ablation in patients are shown in Fig. 4f,g. The end of the clinically used available matt flat-end optical fiber contacted the epicardium vertically to reach maximum penetration depth. Estimating based on the tissue brightness in the top view and cross-section view, the illuminated volume was a hemisphere (Fig. 4f,g (ii, iii)). The light penetration depth by the matt flat-end optical fiber was greater compared to that of iCarP, while the diameter of the hemisphere ( ~ 1.5 cm) was smaller than the diameter of iCarP illuminated myocardium. It can be predicted that light penetration depth would decrease as the gap between myocardium and fiber tip increases, while illuminated area enlarges. As a result, large illumination area and large depth were not achieved simultaneously. In side glow optical fibers, light laterally emits from the 2 cm tip, lowering the peak intensity in the front and broadening the distribution (Supplementary Fig. 7). When vertically inserted into the porcine myocardium, the emitted light illuminated a column volume with a ~ 2 cm diameter and depth equal to 0.5 cm plus fiber insertion length. One can control iCarP illumination area by adjusting patch diameter, to achieve precise treatment and minimize influence on adjacent tissue.

Therefore, when illuminating at similar powers, iCarP is able to evenly distribute the optical energy over a large area compared to clinically used optical fibers and photonic devices with tapered or non-tapered optical fibers in PMCL (without air gap), while achieving satisfactory light penetration depth at the same time.

### Demonstrative application of large-area illumination in tumor treatment

Given the difficulty in fixing the illumination target of flat-end optical fiber on beating hearts, phototherapies of subcutaneous tumors were

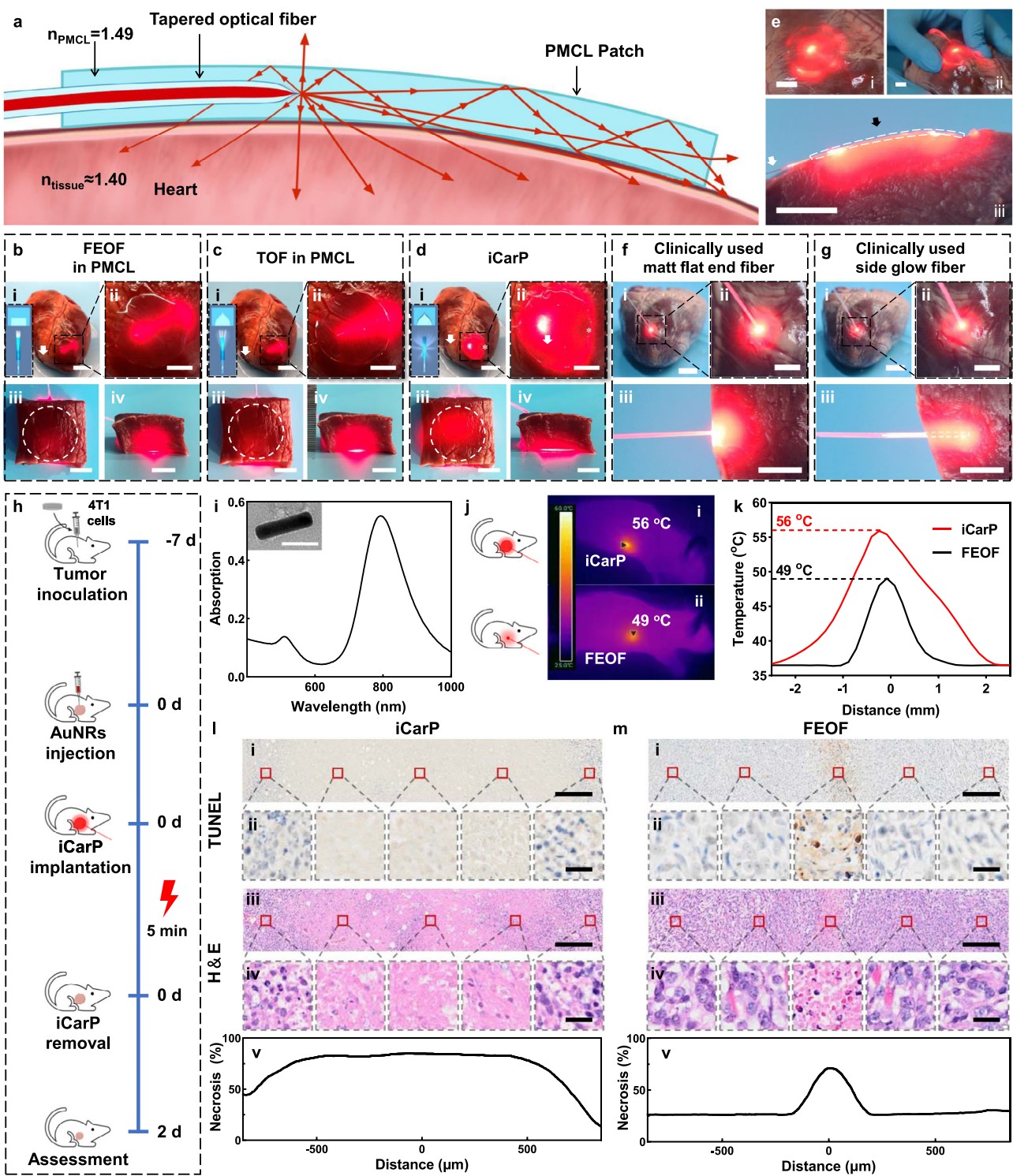

used to evaluate the possible advantages associated with iCarP facilitated large-area illumination. Gold nanorods with maximum absorption at 808 nm were injected into the tumor (~ 5 mm in diameter), and the photothermal effects of illuminations by flat-end optical fiber and iCarP were compared (Fig. 4h, i). The 5 mm diameter iCarP generated a 5 mm cutaneous circular illuminated area, similar to the illuminated area obtained on the hearts. In contrast, the flat-end optical fiber matching the fiber diameter of iCarP concentrated the optical energy in a 2 mm diameter spot, mimicking the illumination of clinically used matt flat-end optical fibers. As iCarP photothermal heating covered the entire tumor, less heat was lost from the skin, which resulted in a larger heated area compared to heating by flat-end optical fiber (Fig. 4j). The

peak temperature was 56 °C, recorded in the center of iCarP heated area, and a 2 mm diameter area was heated to above 45 °C (Fig. 4k). The peak temperature obtained by the flat-end optical fiber was 49 °C, lower than the iCarP counterpart, and only 1 mm diameter area was heated to above 45 °C (Fig. 4k).

Consistently, iCarP photothermal ablation caused greater hyperthermal damage in the tumor. Both TUNEL and H&E staining showed that severe cell apoptosis occurred in the center 2 mm (long axis) hemi-ellipsoid volume of the tumor tissue 2 days after a single 5 min iCarP illumination (Fig. 4l). Less nuclei were found in the photothermally treated area because the cells were dead and failed to maintain the integrity of their nuclei. In comparison, the necrosis

**Fig. 4 | Comparison of illumination scopes and depths, and tumor photothermal ablation effects of different optical fiber/patch photonic devices.**
**a** Cross sectional scheme of the iCarP optical path. **b**–**d** Panorama (i) and magnified images (ii) of ex vivo porcine hearts illuminated by the photonic devices: FEOF in PMCL (**b**), TOF in PMCL (**c**), and iCarP (**d**), insets: optical fiber end structures and corresponding light scattering diagrams, (i) scale bars = 2.5 cm; (ii) scale bar = 1 cm. White arrows indicate the optical fibers. Top view (iii) and front view (iv) of 1.5 cm thick heart tissue illuminated by corresponding photonic devices (devices completely covered by the heart tissue; their locations were indicated by white circles), scale bar = 1 cm. **e** iCarP illumination on ex vivo porcine hearts, scale bar = 1 cm. White arrows indicate the optical fibers, black arrow indicates the location of the air gap. (i) Porcine heart illuminated by iCarP at a lower power compared to that in (**d**), showing the distribution of scattered light. (ii) iCarP illumination on distorted porcine heart. (iii) Cross section of iCarP and porcine myocardium in the same

configuration as in (**a**). **f**, **g** Illumination scopes and depths of clinically used matt flat-end optical fiber (**f**) and side glow optical fiber (**g**). Panorama (i), magnified images (ii) and cross section (iii) of illuminated porcine hearts. Scale bar = 1 cm. **h** Timeline of the animal study. **i** Light absorption spectrum of AuNRs, inset: representative TEM image of > 100 AuNRs. Scale bar = 50 µm. **j** Infrared thermal imaging of mice illuminated by iCarP (i) and FEOF (ii). **k** Temperature of mice illuminated by iCarP (i) and FEOF (ii). **l**, **m** Therapeutic effect of iCarP (**l**) and FEOF (**m**) illumination (representative for *n* = 3 biologically independent samples). (i) Representative TUNEL staining images of illuminated tumor, scale bar = 200 µm. (ii) TUNEL staining images, scale bar = 20 µm. (iii) Representative H&E staining images of illuminated tumor, scale bar = 200 µm. (iv) Magnified H&E staining images, scale bar = 20 µm. (v) Quantitative analysis of necrosis based on H&E staining.

region was a 1 mm diameter column in tumor 2 days after a single illumination by flat-end optical fibers, smaller than the hemi-ellipsoids in iCarP treated tumors (Fig. 4m). The shapes and sizes of the necrosis regions matched the temperature profiles in the tumors treated by iCarPs and flat-end optical fibers, respectively, which demonstrated the applicability of increasing the effects of tumor photothermal therapies by improving illumination area and efficiency via photonic device design.

Similarly, iCarP illumination achieved a more significant tumor growth inhibition in repeated photodynamic therapy compared to illumination by flat-end optical fibers. In iCarp group, the photonic devices indwelled in the implantation site for 3 days, which allowed daily illumination (Supplementary Fig. 6) and showed the potential of iCarP in long-term, repeated phototherapy procedures. FITC (maximum absorbance wavelength 473 nm) was used as the photodynamic agent, together with the gold nanorod (880 nm) described above and Chlorella (660 nm) described below, it demonstrated the compatibility of iCarP with different photosensitizers and corresponding absorbance wavelengths in a broad range. Fourteen days after the first illumination, tumor sizes, and tumor weights in all groups increased, among which iCarP-treated tumors were significantly smaller compared to tumors from other groups (Supplementary Fig. 6). Histological evaluation showed the long-term effects of scattering the optical energy in photodynamic therapy: a higher percentage of necrotic tissue and apoptotic cells in the tumor, and lower density of vessels.

## ICarp supported photosynthesis for myocardial infarction treatment

The tumor phototherapies exhibited the advantages of light scattering by iCarP, particularly in circumstances where disease site coverage by large-area illumination is desired. Theoretically, the horizontal layout of the embedded TOF and the large contact area between iCarP and tissue are desirable for photonic device fixation on targets, particularly moving tissue. Such theoretical advantage of iCarP was evaluated in situ photosynthesis for MI treatment (Fig. 5a). In previous studies, the safety and efficacy of algae-based photosynthesis systems have been demonstrated in treating hypoxia diseases[43,44]. The infarcted rat myocardium turned from pale white to light green after intramyocardial Chlorella injection, showing that the injected Chlorella distributed throughout the infarct (Supplementary Fig. 8). No major bleeding occurred during or after Chlorella injection as shown in Supplementary Fig. 8 and Supplementary Movie 4. Limited by the anesthesia time, Chlorella was continuously illuminated for 3 h after injection in a closed chest state (continuous illumination could theoretically be extended if allowed). The illumination intensity and scope were stable in the meantime, as evaluated by the light penetrated rat chest. After illumination, the TOF was twisted, pulled to detach from the PMCL substrate as chest remained closed (Fig. 5b and Supplementary Movie 5). TOF surface was fluorinated to controllably weaken the interaction between the TOF and PMCL substrate, which facilitates

TOF removal and the production of the air gap. The biodegradable PMCL substrate remained adhered to the epicardium, exhibiting no detachment (Fig. 5b). iCarP did not adhere to the chest, as the reactive groups of CCS@gel were consumed during crosslinking and covalent bonding with iCarP and epicardium (Supplementary Movie 6). The structures of both TOF and PMCL substrates were intact. The firm adherence between the PMCL substrate and the epicardium is an important basis for repeated illumination and extended mechanical support after removing the optical fiber.

Cardiac cell apoptosis 3 days post-MI was detected by TUNEL staining (Fig. 5c). The ischemic injury caused significant cell death in the MI group compared to the Sham control. The percentage of apoptotic cardiac cells significantly decreased in iCarP + /Light- group compared to the MI group, showing that iCarP alone can inhibit cardiac cell death by providing mechanical support (Fig. 5d). Between the two iCarP treated groups, the iCarP + /Light+ group showed a lower percentage of apoptotic cells compared to iCarP + /Light- group (Fig. 5d), attributed to oxygen produced by light triggered photosynthesis. The cell densities in the infarcted myocardium of MI and two treatment groups were higher compared to Sham myocardium, which was due to recruitment of inflammatory cells in the infarcts. Immunofluorescent staining of apoptosis related molecules Bcl-2, Bax, and Cleaved Caspase-3 exhibited consistent results. As shown in Supplementary Fig. 9, expression level of anti-apoptotic Bcl-2 was reduced by MI compared to that in Sham group, and was highest in iCarP + /Light+ group. In addition, the expression levels of pro-apoptotic Bax and Cleaved Caspase-3 were lowest in iCarP + /Light+ group, significantly lower compared to them of MI group. RNA-seq showed that the expression levels of representative pro-apoptosis genes in LV myocardium, including *Bax*, *Htra2* and *Myc* were elevated by MI compared to baseline levels in Sham rats, and were decreased by in situ photosynthesis in iCarP + /Light+ group (although showing the trend, significant difference in BAX level between MI and iCarP + /Light+ groups was not detected, which is probably due to the difference in post-transcriptional regulation or systemic errors of the RNA-seq and immunofluorescent staining techniques). In vitro cell culture showed that illumination (660 nm, 55 mW, 3 h) with or without Chlorella did not induce intracellular ROS accumulation in cardiomyocytes, as shown in Supplementary Fig. 11. Consecutive illumination for 12 h did not decrease the viability of cardiomyocytes in vitro, and 6 h consecutive illumination did not cause damage to rat hearts in vivo, indicating the safety of extended iCarP illumination.

Myocardial fibrosis detected by Masson's trichrome staining 28 days post-surgery (Fig. 5e) exhibited a trend same to apoptosis results, which is expected as fibrosis is considered a result of cardiac damage[45]. Compared to MI group (25.6 ± 2.8%), iCarP + /Light- treatment significantly reduced cardiac fibrosis (22.1 ± 2.0%), while iCarP + / Light+ group had the lowest fibrosis level (20.1 ± 2.5%). Muscle tissue can be found stained in red in infarcted left ventricle from iCarP + / Light+ treated hearts, whereas the entire infarcted LV was fibrotic in

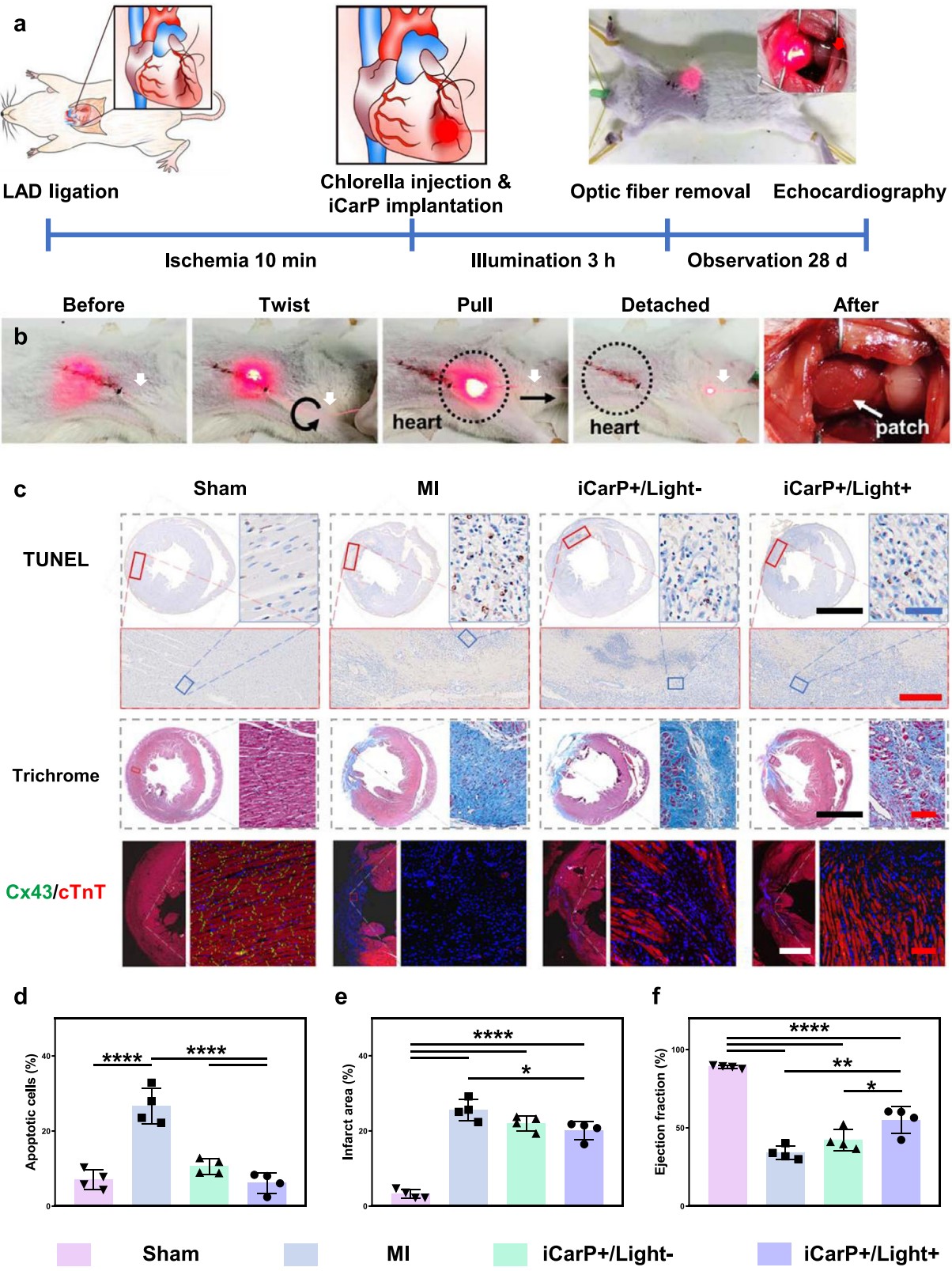

hearts from MI group (Fig. 5c). The expression levels of representative fibrosis genes, including *Tgfb1* and *Tgfb2* for TGF-β signaling pathway and *Col5a3* for collagen fibril organization were elevated in MI group compared to Sham group, and decreased in iCarP + /Light+ group, supporting the effects of iCarP triggered in situ photosynthesis on fibrosis in reducing fibrosis (Supplementary Fig. 10). In vitro cell culture showed that none of Chlorella, iCarP illumination, or their

combination significantly changed the viability or proliferation rate of primary fibroblasts (Supplementary Fig. 12), again indicating inhibiting cardiac cell apoptosis may be the primary reason for observed reduction in fibrosis. Left ventricle ejection fraction (LVEF) significantly decreased in MI group compared to Sham, indicating cardiac dysfunction after myocardial infarction. Although there was an increasing trend, LVEF of iCarP + /Light- group (with theoretical

**Fig. 5 | Myocardial infarction treatment with iCarP illumination induced in situ photosynthesis. a** Timeline of the animal study. Sham group: rats underwent thoracotomy only without LAD ligation; MI group: rat underwent LAD ligation, without treatment; iCarP + /Light- group: MI rats received chlorella injection and iCarP implantation, without illumination; iCarP + /Light+ group: MI rats received chlorella injection and iCarP illumination. **b** Removal of TOF after treatment while chest is closed. **c** Histological evaluation of therapeutic outcome by in situ photosynthesis: representative TUNEL staining images of hearts sections 3 d after MI (apoptosis); representative Masson's trichrome staining of rat hearts 28 d after MI (fibrosis); representative immunofluorescence staining images of Cx43 (green),

cTnT (red), and nuclei (blue) in LV 28 d after MI (down). Scale bars in TUNEL staining images: black = 5 mm, red = 500 μm, blue = 50 μm, scale bars in other staining images: black = 5 mm, red = 100 μm, white = 2 mm. **d–f** Quantitative analysis of (**d**) apoptotic cells, (**e**) infarct area, and (**f**) ejection fraction based on TUNEL staining, Masson's trichrome staining, and day 28 echocardiography, respectively (*n* = 4 per group). Statistical significance was calculated using one-way ANOVA with Tukey's posttest and data are presented as means ± SD. $*p < 0.05$, $**p < 0.01$, $***p < 0.001$, $****p < 0.0001$. Source data and exact *p*-values are provided in the Source Data file.

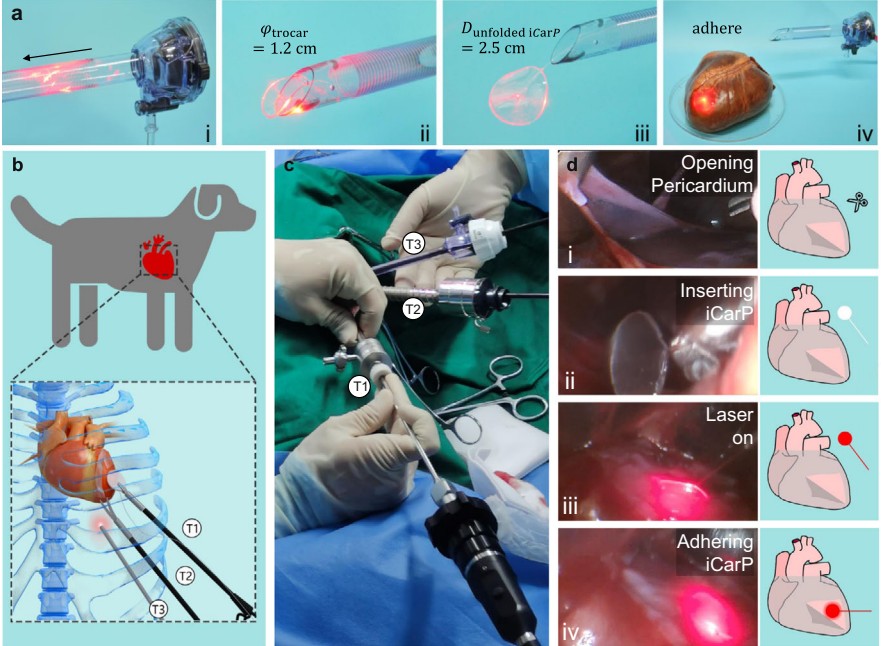

**Fig. 6 | ICarP is compatible with minimally invasive implantation. a** Passage of iCarP at on-state through trocar: (i) folded iCarP (2.5 cm diameter) inserted into trocar (1.2 cm diameter), (ii) iCarP passing through the exit of trocar, (iii) an iCarP passed through trocar and unfolded, (iv) the unfolded iCarP was adhered onto the surface of porcine heart. **b** Schematic illustration of the minimally invasive

implantation of iCarP in a canine model. T1: trocar 1 for grasping forceps; T2 trocar 2 for curved scissors; T3: trocar 3 for endoscope. **c** ICarP implantation under thoracoscopy in dogs without open-heart surgery. **d** Illustration of the steps of iCarP implantation: (i) opening pericardium; (ii) inserting iCarP; (iii) turning on illumination; (iv) adhering iCarP.

mechanical support) was not significantly higher compared to LVEF of MI group, despite the short-term cell protection effect shown in TUNEL staining. ICarP + /Light+ treatment significantly increased LVEF compared to MI and iCarP + /Light- group, demonstrating the therapeutic effect of in situ photosynthesis (Fig. 5f).

Expression of gap junction protein Cx43 and cTnT, a key regulator of myocardial muscle contraction in ventricular myocytes indicates the condition of LV myocardium. Immunohistochemical staining 28 days post-MI showed a significantly lower Cx43 density and tissue level discontinuity in cTnT expression, indicating a severe damage in infarcted myocardium from MI group (Fig. 5c). The expression level and organization of cTnT and Cx43 in infarcted myocardium improved significantly after being treated by iCarP + /Light+ treatment (Fig. 5c), indicating a healthier myocardium supported by local oxygen supply. The effects of iCarP on myocardial electrophysiology and myocardial contractility in vitro were evaluated by multifunctional hybrid integrated cardiomyocyte biosensor[46], as shown in Supplementary Fig. 13. Results showed that illumination and Chlorella did not affect cardiomyocyte contraction and relaxation, but the firing rate was higher with Chlorella, which is believed to be a result of increased energy supply from photosynthesis.

### Minimally invasive implantation in canine model

The compatibility of iCarP with minimally invasive implantation was demonstrated in an in vitro model and a canine thoracoscopic

implantation surgery (Fig. 6b). Attributed to the flexibility of PMCL, iCarP with a diameter relevant to human hearts (2.5 cm) can be bent and transported through a 1.2 cm diameter Trocar sheath (which provides access ports in a thoracoscopy) with small sliding friction (Fig. 6a). The laser was on during the whole iCarP advancing process inside the sheath, no interruption or significant intensity change was observed. After protruding the Trocar sheath channel, the iCarP device unfolded and recovered its original, flat geometry (Fig. 6a, iii). Device illumination was stable and continuous in the shape recovery stage, as well. Using the same adhesive in the rat experiments and illumination demonstrations, the unfolded iCarP was firmly attached to the epicardium of an ex vivo porcine heart (Fig. 6a, iv). Illumination performance in terms of scope, intensity, and stability was not impaired by the simulative ex vivo implantation, indicating that the structural integrity, particularly the air gap between the TOF and PMCL was functionally maintained.

In the canine thoracoscopic implantation surgery, three access channels were successfully created by Trocar sets (Fig. 6b, c). The anterior wall of LV was exposed first by opening the pericardium (Fig. 6d, i). Under video guidance, bent iCarP were delivered into the thoracic cavity through a 1.2 cm Trocar sheath with a tissue forceps. Similar to the ex vivo Trocar delivery demonstration, bent iCarP passed through the Trocar sheath with little resistance, and recovered to its original, flat form, as shown in Fig. 6d (ii). The structural basis of the

waveguiding and light scattering function was well maintained, as the scattered light was collected by the inserted camera (Fig. 6d,iii). It is worth noting that the camera was placed behind the PMCL patch, which is a solid evidence showing that the air gap and light reflection inside the PMCL substrate supported propagating optical energy in the rearward direction in vivo, in consistency with the simulation results and ex vivo demonstration. The attached iCarP moved along with the epicardium, showing no detachment or sliding, despite the significantly greater amplitude of heart contraction. The illumination stayed stable and evenly distributed in the patch-covered area during the entire experiment (Fig. 6d, iv). The influence of iCarP illumination on heart rate of healthy and MI dogs and rats was evaluated. As shown in Supplementary Fig. 14 and Supplementary Movie 7, iCarP illumination did not affect the heart rate of the dog, or cause arrythmia. These results showed the adaptability and safety of iCarP with minimally invasive implantation surgeries on other internal organs and tissues.

## Discussion

For light-based diagnosis, therapy, and surgery applications, a wide spectrum of choice and precise control over illumination scope, intensity, wavelength, duration, and depth are desirable in local light transmission to internal organs[15,25,47]. Direct irradiation from organ surface fixed implantable optical devices is expected to better meet the requirements for targeted delivery of sufficient optical energy. In this study, we constructed a flexible light-scattering optic fiber patch, iCarP for internal organ and tissue illumination, and demonstrated its therapeutic function in treating rat myocardial infarction. The structure of the tapered optical fiber, the air gap between the optical fiber and PMCL patch, and the highly refractive PMCL integrated into iCarP design collectively scatter and guide optical energy to tissue at the implantation site to achieve large area and deep illumination on internal organs and tissues.

The air gap is the key design in our iCarP device to pivot light transmitting along the optical fiber to target tissue, such a structural feature could be widely introduced to enlarge the divergence angle for optical fiber-based photonic devices while keeping the advantages of high power, strong interference-resistance, and maneuverability. The essence of the strategy is creating a micrometer scale space with low refractive index between the two substances with similar, higher refractive indexes (the TOF and the patch substrate). Our results have demonstrated that the low refractive index substrate in the gap between the TOF and PMCL is crucial for light scattering. Air has a refractive index ($n_{air} = 1.0003$) closer to that of vacuum ($n_{vacuum} = 1$) compared to commonly used biomaterials, thus an air gap theoretically has higher scattering performances compared to a gap of filler or coating material. Fabricating the air gap by pulling the TOF with a customized instrument could be both convenient and precise. In addition, this method is expected to be almost universally compatible with different patch materials, as the fluorinated coating provides lubrication, hence lowering the risk of damaging the patch material or increasing the roughness of the air/patch interface. The shape of the air gap was controlled by the tapering process, a technique commonly used in the field of communication but has not been employed in photonic devices for biomedical applications. This technique is compatible with multimodal optical fibers with different diameters and core/cladding configurations.

Besides light scattering, other design features contribute to internal organ and tissue illumination. In addition to light scattering, the patch substrate protects the fragile, micrometer-scale TOF tip from the mechanical stress by tissue, isolates the air gap from contamination by proteins and cells, and firmly fixes the optical fiber at the implantation site, ensuring stable, precise illumination. The multimodal optical fiber used in our design supports transmission of visible light and infrared light, thus can deliver light within the spectrum with low tissue penetration which can hardly reach internal tissues from ex vivo light sources[24]. The iCarP design principle is theoretically compatible with optical fibers and photonic patches that deliver lights with wavelengths <400 nm. For light in the tissue penetration window, the optical fiber iCarP could transmit light with significantly higher intensity and spatial precision, as there is no energy dissipation in the tissue between the target and light source given that iCarP is directly attached to the target tissue.

ICarP is suitable for creating microenvironments in internal organs and tissues with controllable illumination. In addition to the capability of long-term, repeated illumination on internal organs and tissues, demonstrated in this work, one could theoretically use a portable, programmable miniature light source to input optical energy, which could avoid significantly impairing the movability of patients. In addition to photosynthesis, light delivered by iCarP could potentially be employed in optogenetics, photothermal therapy, photo-sensitive drug release and targeting, and light-based sensing/diagnosis applications. The photonic device could not only deliver optical energy, but also be a carrier for information, to develop biomedical applications integrating medical data communication[48–50]. Optical energy and information could go in both direction in the optical fiber, thus can potentially integrate therapy and diagnosis in a single closed-loop device.

As iCarP is composed of a degradable patch and a removable optical fiber, the device shall leave no toxic or non-degradable residues after temporary services. Since no circuits is integrated in iCarP, no Joule heat is generated in the device, thus the risk of thermal injury is negligible, not only in the illuminated tissue but also in adjacent tissues along the TOF/light path. In our design, the diameter of the flexible optical fiber is only 100 μm, which is favorable for limiting the risk of providing a passage for ambient bacteria to enter the thoracic cavity.

Despite the promising results with the iCarP based strategy for the illumination of internal organs and tissues, this study is still associated with the following limitations. First, the long-term foreign body responses of iCarP were not thoroughly investigated. The degradation of PMCL is relatively slow, adhesion of the device to the thoracic wall was observed in the experimental animals. Second, limited by anesthesia time of rats, illumination time was limited to 3 h per round. Performance and influence of extended, continuous iCarP illumination need to be studied to evaluate the safety and efficacy of the illumination strategy. Third, light scattered from both sides of the iCarP device, illuminating the tissue on the opposite side of the target tissue. A coating that could completely reflect the light would be favorable in applications optical energy is desired to be strictly guided to target tissue in order to avoid side effects.

In summary, we presented a photonic device-based strategy for large area, deep, noninvasive illumination on internal organs and tissues. A tapered optical fiber, and more importantly, a micrometer sale air gap between the TOF and a flexible, highly refractive, and biodegradable polyester were integrated into the photonic device, iCarP, which scatters and pivots forward going light into lateral directions. ICarP supports long-term, wide spectrum, continuous/pulsatile, deep penetrating (1.5 cm) illumination on beating animal hearts, without physically damaging the target tissue. In vivo photosynthesis as a potential application and the compatibility with minimally invasive implantation in large animal models were demonstrated. Our study showed that iCarP could be a safe and powerful device suitable for internal organs/tissues illumination and associated therapy and diagnosis.

## Methods

### Research compliances

All animal experiments were approved by the Guidelines of Animal Care and Use Committees of Zhejiang University (ZJU20210164, ZJU20220008) and Zhejiang Academy of Medical Sciences (ZJCLA-IACUC-20010027, ZJCLA-IACUC-20120002). Porcine hearts in this

work were purchased from supermarkets for demonstration. In order to avoid experimental differences caused by animal sex, animals of the same sex were used in the same experiment. The maximal tumor size in this study was less than 1000 mm³, which was far less than maximal tumor size (2000 mm³) permitted by Zhejiang Academy of Medical Sciences.

## Materials

Acryloyl chloride (Sigma-Aldrich, USA), triethylamine (Sigma-Aldrich, USA), Rhodamine B (Sigma-Aldrich, USA), lipase B Candida antarctica (recombinant from Aspergillus oryzae, Sigma-Aldrich, USA), I2959 (BASF SE, Germany), Sn(Oct)$_2$ (Alfa Aesar, UK), gold nanorods (AuNRs) (Beijing Zhongkeleiming Daojin Technology, China), FITC (Yeasen Biotechnology, China) and Chlorella vulgaris (Guangyu Biological Technology, China) were used as received. Dulbecco's modified Eagle's medium (DMEM) (Gibco, USA), fetal bovine serum (Sijiqing, China), Cell counting kit-8 (CCK-8) and dichloro-dihydro-fluorescein diacetate (DCFH-DA) (Beyotime Biotechnology, China), One Step TUNEL Apoptosis Assay Kit (Beyotime Biotechnology, China), Anti-Connexin 43/GJA1 antibody-Intercellular Junction Marker (Acbam, UK, ab11370, Polyclonal, 1:2000 dilution), Anti-Cardiac Troponin T antibody (Acbam, UK, ab209813, EPR20266, 1:4000 dilution), Anti-Bcl-2 antibody (Abcepta, China, P10415, Polyclonal, 1:100 dilution), Anti-Bax antibody (Abcam, UK, ab32503, E63, 1:250 dilution), Cleaved Caspase-3 Rabbit mAb(Cell Signaling Technology, USA, #9664, 1:2000 dilution), Anti-IL6 antibodies (Affinity biosciences, USA, DF6087, Polyclonal, 1:200 dilution), Anti-CD31 antibody (Abcam, UK, ab182981, EPR17259, 1:2000 dilution) and Anti-TNF-α Polyclonal antibody (ImmunoWay Biotechnology Company, USA, YT4689, Polyclonal, 1:200 dilution) were used according to corresponding protocols. 4T1 cell line (CRL-2539) and H9C2 cell line (GNR 5) were purchased from ATCC and Cell Bank of Typical Culture Collection of Chinese Academy of Sciences, respectively. 1,4-benzenedimethanol (Aladdin, China) was dried under reduced pressure at 80 °C before use. 4-methyl-ε-caprolactone was synthesized by Baeyer-Villiger oxidation of 4-methylcyclohexanone according to the literature and distilled from CaH$_2$ under vacuum pressure before use[51]. Other chemicals were purchased from Sinopharm Chemical Reagent, China, and used as received. Indwelling needle (Berran, Germany), absorbable suture (Shanghai Pudong Jinhuan Medical Supply, China), clinically used side glow optical fiber, and matt flat-end optical fiber (Nanjing Chunhui Technology Industry, China), lidocaine hydrochloride (Shanxi Taiyuan Pharmaceutical Industry, China), propofol (Guangdong Jiabo Pharmaceutical, China), isoflurane (Tianjin Rep Biopharmaceutical, China), atropine Sulfate (Shanxi Ruicheng Kelong Veterinary Drug, China), ringer lactate (Jisheng Pharmaceutical, China), ceftiof (Inner Mongolia Federal Pharmaceutical, China) and meloxicam (Labiana Life Sciences, Spain) were used as received.

## Polymer synthesis

Di-acrylated poly(4-methyl-ε-caprolactone) (PMCL) was synthesized as previously described (Supplementary Fig. 1)[52]. Briefly, PMCL diol was synthesized by ring-opening polymerization of 4-methyl-ε-caprolactone (MeCL), using 1,4-benzenedimethanol as initiator and tin (II) ethylhexanoate (Sn(Oct)$_2$) as catalyst. Then, end diol groups were acrylated with excess acryloyl chloride and triethylamine. CCS@gel was prepared as previously described (Supplementary Fig. 2)[53].

## Photo-crosslinking

Photoinitiator I2959 (0.01 g) was dissolved in 0.5 mL of dichloromethane (DCM), and mixed in dark with 2 mL di-acrylated PMCL. The mixture was poured into polytetrafluoroethylene (PTFE) molds and dried in a vacuum oven to remove DCM. Di-acrylated PMCL was crosslinked by 365 nm UV irradiation of 500 mW/cm$^{-2}$ for 10 min.

## Refractive index of crosslinked PMCL

Refractive index ($n_{PMCL}$) of PMCL films was measured by an WYA-Z Abbe refractometer (Shanghai INESA Physico Optical Instrument, China). By sandwiching the 250 μm thick PMCL films in the illuminating prism and refracting prism, and adjusting the scale knobs, $n_{PMCL} = 1.49$ was obtained.

## Device fabrication

Optical fibers were drawn with the Fujikura fusion splicer 80 S+ under [MM-MM/Taper splice] splice mode (Fig. 2a). Elongation length was set at 400 μm. Tapered optical fibers (TOFs) with a length of tapered region of 280 μm were obtained. The diameter of the TOF tip is 1 μm, and the shape of the TOFs can be described as Supplementary Eqn. 1, in which F(z) is the radius of the fiber tip, z is the distance to fiber tip. TOFs were then hydroxylated in a plasma cleaner, and fluorinated in a vacuum desiccator with 100 μL trichloro(1H,1H,2H,2H-tridecafluoro-n-octyl)silane for 12 h with a pressure <0.09 MPa.

To integrate TOFs with PMCL, TOFs were fixed on a glass slide, and their tips were immersed in di-acrylated PMCL drops (Fig. 2a). A cover glass slide was placed on top to reduce the contact of PMCL with oxygen and ensure that the thickness of the PMCL layer was equal to the diameter of standard optical fibers. After UV crosslinking, excess PMCL was removed to obtain round, transparent cardiac patches, which were peeled off from the glass slide. By gently pulling the TOF, an air gap whose geometry can be easily controlled by the distance between the TOF tip and air gap tip (d) was created between the PMCL substrate and the TOF, which was expected to increase the light divergence angle.

Follow the above operation and remove the step of pulling optical fiber, we fabricated the iCarP based on TOF without air gap. Then, TOF was replaced by flat-end optical fiber (FEOF) to fabricate iCarP based on flat-end optical fiber without air gap.

## Device simulation and characterization

Beam Propagation Method (BPM) in commercial photonic simulation software RSoft was used to simulate the light scattering capability of the TOF surrounded by different combinations of air and PMCL, marked by the divergence angle of emitting light. The geometry of TOF was measured from microscope images and reconstructed in MATLAB. Refractive indexes of 1.49, 1.4613, and 1.4562 were assigned to the surrounding PMCL, the core of TOF, and the cladding of TOF, respectively. The transmission mode in TOF was $LP_{01}$.

A fluorescent solution was used to visualize the light path. The 250 μL Rhodamine B in ethanolic solution with a concentration of $8.35 \times 10^{-3}$ mol/L was dispersed in 50 mL 52% glycerol-water solution with a refractive index of 1.40, which is close to that of tissues. Devices carrying 520 nm laser were immersed in the solution. As the emergent laser-excited fluorescence, images were taken to record the light scattering paths.

ICarP and sensor for dissolved oxygen (Unisense, Denmark) were immersed into 40 ml Chlorella suspension ($2 \times 10^7$/mL) in a centrifuge tube, sealed and wrapped with tin foil to prevent oxygen leakage and block ambient light. ICarP was connected with a laser diode (FCL.660.F100.MM) with 660 nm wavelength and 55 mW output power. Concentration of dissolved oxygen in Chlorella suspension was recorded.

## Illumination demonstration in live rats and on excised porcine hearts

These animal experiments were approved by the Guidelines of Animal Care and Use Committees of Zhejiang University (ZJU20210164) and Zhejiang Academy of Medical Sciences (ZJCLA-IACUC-20010027). Male SD rats (6 weeks old, 180–220 g) were purchased from Zhejiang Academy of Medical Sciences and used in this study. Rats were anesthetized with 1% pentobarbital (40 mg/kg) by intraperitoneal injection,

followed by endotracheal intubation and assisted ventilation. Subsequently, rats were placed in the supine position, followed by a left thoracotomy and pericardectomy to expose the heart. ICarP was adhered to the epicardium of the infarct area by CCS@gel (Supplementary Fig. 2).

To demonstrate the versatility of iCarP to transmit light at different modes, laser diodes with wavelengths of 405 nm (140501-55), 520 nm (LP520-SF15), 660 nm (LP660-SF20) from Thorlabs, Inc. were used as light sources, output power of laser diode or illumination intermittent were adjusted by controlling the pump current of laser diode drivers. Illumination on rat hearts was recorded by the camera.

The iCarP was adhered onto the porcine heart ex vivo to observe the illumination area. The iCarP was adhered onto the dissected porcine heart ex vivo to observe the illumination depth at a view of longitudinal section. The porcine heart was cut into cuboids of 1.5 cm thicknesses where iCarP was completely covered underneath, turn on the light (75 mW) and observe the penetration depth.

After illumination, TOFs were firstly twisted by hand to loosen the interfacial integration between TOFs and PMCL substrate, when no more resistance was felt, TOFs were quickly pulled out.

### Tumor photothermal therapy in mice

This animal experiment was approved by the Guidelines of Animal Care and Use Committees of Zhejiang Academy of Medical Sciences (ZJCLA-IACUC-20120002). Female BALB/c nude mice (12 weeks old) were inoculated with $1 \times 10^6$ 4T1 cells into the breast pad of the mice and randomly divided into six groups: (1) Control; (2) AuNRs; (3) flat-end optical fiber; (4) iCarP; (5) AuNRs + FEOF; (6) AuNRs + iCarP. For groups 2, 5, and 6, 50 μL of gold nanorods (0.1 mg/mL) were injected into the subcutaneous tumor. A solid-state laser with a wavelength of 808 nm (DE51543, Changchun New Industries Optoelectronics Tech, China) was used as the light source (300 mW). Tumor temperature was measured by a thermal imager. Forty-eight hours after treatment, tumor tissues were excised from mice. Tissue sections in each group with 5 μm thickness were stained with hematoxylin and eosin (H&E) and TUNEL. All sections were examined by virtual slide microscopy (Olympus VS200, USA).

### Rat myocardial infarction treatment

This animal experiment was approved by the Guidelines of Animal Care and Use Committees of Zhejiang University (ZJU20210164) and Zhejiang Academy of Medical Sciences (ZJCLA-IACUC-20010027). Male SD rats were anesthetized with 4% chloral hydrate by intraperitoneal injection, followed by endotracheal intubation and assisted ventilation. The rats were placed in the supine position, followed by a left thoracotomy and pericardectomy to expose the hearts. Then the left anterior descending (LAD) coronary artery was ligated with a 6−0 silk suture at approximately 2–3 mm from its origin between the left atrium and the pulmonary artery conus to create left ventricular (LV) infarction.

Rats were divided into 4 groups in random: (1) Sham group (underwent thoracotomy, without LAD ligation); (2) MI group (without iCarP); (3) iCarP + /Light- group (iCarP without illumination); (4) iCarP + /Light+ group (iCarP with illumination). 10 min after LAD ligation, Chlorella (100 uL, $2 \times 10^7$/mL) suspension was injected into infarcted LV of rats in iCarP groups, followed by swab pressing for hemostasis, and iCarP adhesion via CCS@gel on the epicardium of the infarct area. Subsequent to iCarP implantation, chest cavity, muscles, and skin were sutured with 3−0 silk sutures. Rat hearts in iCarP + /Light + were illuminated continuously for 3 h (55 mW), no illumination was performed in other groups.

Cardiac functions at 7 d and 28 d post MI were assessed using echocardiography (VisualSonics, Canada), M-mode echocardiographic and two-dimensional images in a parasternal short and long axis were recorded. Left ventricular ejection fraction (LVEF), left ventricular fractional shortening (LVFS), left ventricular end-diastolic volume (EDV), left ventricular end-systolic volume (ESV) were calculated as previously described[54].

Rats were anesthetized and sacrificed at 7 and 28 d post MI to harvest short-term and long-term treated hearts, respectively. Rat hearts were fixed with 4% paraformaldehyde, embedded in paraffin, and sectioned. The short-term hearts were TUNEL stained to assess cell apoptosis. Masson's trichrome staining was applied on long-term cardiac tissues to evaluate LV geometry and fibrosis. Gap junction protein Cx43, cardiac troponin T (cTnT), and DAPI were stained to evaluate the microscopic morphology of the myocardium. Stained slides were observed using an Olympus IX51 microscope. Images were captured using DP2-BSW software (Olympus, VS200, USA), and analyzed by ImageJ software.

### Minimally invasive implantation

These animal experiments were approved by the Guidelines of Animal Care and Use Committees of Zhejiang University (ZJU20220008). This animal experiment was approved by the Guidelines of Animal Care and Use Committees of Zhejiang University (ZJU20220008). Male Beagles (2 years old, 13 kg) were fasted for 8 h before surgery to prevent reflux from blocking the esophagus and trachea during anesthesia. Surgical instruments and aseptic hole-towel were sterilized under high temperature or in the benzalkonium bromide diluent. An intravenous indwelling needle was embedded in the forearm, and treated by diluted heparin. The anesthesia machine was confirmed free of air leakage and obstruction of oxygen and isoflurane flows. The endotracheal intubation balloon was also examined for air leakage and disinfected by benzalkonium bromide diluent.

Anesthesia of dog was induced by propofol, while heart rate and body temperature were recorded. Dog mouth was opened, followed by pulling out the tongue with gauze, and laryngoscope-assisted intubation. Afterward, the anesthesia machine was connected to allow the dog to quickly enter the surgical anesthesia period and fix the tracheal intubation. Physical conditions were checked every five minutes. On the ventilator, tidal volume was set to 75 mL, and the breathing ratio was 1:2. Multi-point subcutaneous lidocaine injection in the operation area was employed for local anesthesia.

Ceftiofur solution was used for anti-infection. Lactated Ringer was used to replenish body fluids. The dog was adjusted to a supine posture on the heated operation table, the thoracic surgery area was shaved, cleaned, disinfected, and aseptic hole-towel was laid. The access paths of the three Trocars were between the eighth and ninth ribs, the ninth and ten ribs, the ten and eleventh ribs, respectively. A 1 cm incision was created on the skin at the first position, and the subcutaneous tissue was bluntly separated, then the Trocar was slowly pushed into the chest cavity. After removing the puncture needle, endoscope was accessed through the above channel to observe the potential positions and depths of the second and third entry points. After the three channels were established, surgical devices were accessed in sequence, pericardium was grabbed by grasping forceps and cut by blunt bending scissors, then the scissors were substituted by iCarP for attachment. ICarP illumination (75 mW) was tested, and all procedure was observed through the endoscope. Subsequently, the chest cavity was closed, and iodophor was used to prevent infection. The ventilator was retained for 1 h before recovery of spontaneous respiration. For analgesia and anti-inflammation, 0.6 mL of meloxicam was subcutaneously injected.

### Statistics and reproducibility

Statistical analysis was performed using GPower and Prism. Results are shown as mean ± SD unless stated otherwise. T-tests were used to compare results from two groups. One-way analysis of variance (ANOVA) was used for multiple comparisons unless stated otherwise.

*p*-value of <0.05 was considered statistically significant. All in vitro and in vivo results are representative of two to six independents.

## Reporting summary

Further information on research design is available in the Nature Portfolio Reporting Summary linked to this article.

## Data availability

RNA-Seq data were deposited at National Center for Biotechnology Information Gene Expression Omnibus (GEO) with accession number "GSE223691". All other relevant data supporting the key findings of this study are available within the article and its Supplementary Information files or from the corresponding author upon reasonable request. Source data are provided with this paper.

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

## Acknowledgments

This study is financially supported by the National Key Research and Development Program of China (no. 2019YFE0117400), National Natural Science Foundation of China (no. 82202328, 32000971, 61975173, 32250008, 62271443, 81971667), Major Scientific Research Project of Zhejiang Lab (no. 2019MC0AD01), Key Research and Development Project of Zhejiang Province (no. 2021C05003), and Binjiang Institute of Zhejiang University (ZY202205SMKY007). The authors thank research staff Qiaojuan Shi from Zhejiang Academy of Medical Sciences for her assistance in animal studies, and Yuhang Tao from Sir Run Run Shaw Hospital for his assistance with electrocardiogram monitoring and analysis.

## Author contributions

Y. Zhu designed the overall project. K. Deng and Y. Tang fabricated and characterized the photonic devices, including simulation. D. Zhong conducted Chlorella-related experiments. H. Zhang performed the minimally invasive implantation. W. Fang and Q. Hu designed and prepared the tissue adhesive. L. Shen, J. Pan, K. Deng, and H. Wang performed the small animal study and assisted in the large animal study. Y. Lu and C. Gao assisted with the device fabrication. C. Chen, H. Wan, LJ. Zhuang, P. Wang, and T. Ren performed in vitro cardiomyocyte electrophysiology and contractility experiments. Y. Gao, Q. Jin, LN. Zhuang and J. Zhai performed RNA-seq analysis. Y. Xiao, Z. Wang, Y. Zhu, and M. Lang designed and synthesized PMCL. Y. Zhu and L. Zhang designed the photonic devices. M. Zhou, Y. Zhang, and H. Wang designed the animal studies. Y. Zhu, L. Zhang, M. Zhou, K. Deng, Y. Tang, D. Zhong, C. Gao and H. Zhang wrote the manuscript.

## Competing interests

Y. Zhu, L. Zhang, Y. Tang, K. Deng, and L. Shen have a granted patent (CN 114137663B) related to the present study and thus may have related financial interests. All other authors declare no competing interests.

## Additional information

[1]MOE Key Laboratory of Macromolecular Synthesis and Functionalization, Department of Polymer Science and Engineering, Zhejiang University, Hangzhou 310027, China. [2]Research Center for Humanoid Sensing, Zhejiang Lab, Hangzhou 311100, China. [3]State Key Laboratory of Modern Optical Instrumentation, College of Optical Science and Engineering, Zhejiang University, Hangzhou 310027, China. [4]Shanghai Key Laboratory of Advanced Polymeric Materials, Key Laboratory for Ultrafine Materials of Ministry of Education, School of Materials Science and Engineering, East China University of Science and Technology, Shanghai 200237, China. [5]Zhejiang University-University of Edinburgh Institute (ZJU-UoE Institute), School of Medicine, Zhejiang University, Haining 314400, China. [6]College of Animal Sciences, Zhejiang University, Hangzhou 310058, China. [7]Biosensor National Special Laboratory, Key Laboratory for Biomedical Engineering of Education Ministry, Department of Biomedical Engineering, Zhejiang University, Hangzhou 310027, China. [8]Institute of Plant Quarantine,

Chinese Academy of Inspection and Quarantine, Beijing 100176, China. [9]Department of Cardiology, Cardiovascular Key Laboratory of Zhejiang Province, Second Affiliated Hospital, Zhejiang University, Hangzhou 310009, China. [10]San Francisco Veterans Affairs Medical Center, San Francisco 94121, USA. [11]Institute of Translational Medicine, Zhejiang University, Hangzhou 310009, China. [12]Key Laboratory of Cancer Prevention and Intervention, National Ministry of Education, Zhejiang University, Hangzhou 310009, China. [13]Dr. Li Dak Sum & Yip Yio Chin Center for Stem Cell and Regenerative Medicine, Zhejiang University, Hangzhou 310058, China. [14]Binjiang Institute of Zhejiang University, Hangzhou 310053, China. [15]Key Laboratory of Cardiovascular Intervention and Regenerative Medicine of Zhejiang Province, Sir Run Run Shaw Hospital, Zhejiang University, Hangzhou 310016, China. [16]These authors contributed equally: Kaicheng Deng, Yao Tang, Yan Xiao, Danni Zhong. ✉e-mail: zhoum@zju.edu.cn; cygao@zju.edu.cn; zhang_lei@zju.edu.cn; zhuyang@zju.edu.cn

