## [Peer Review File · Nature Communications]

REVIEWER COMMENTS

Reviewer #1 (Remarks to the Author):

In this work, the authors presented an iCarP device, which could achieve internal organ illumination. The optical fiber technology upgradation may have some novelty, but most of other findings does not have enough novelty or innovative aspects, for example materials or mechanisms, to be published in this top journal. Therefore, I would suggest this manuscript to be transferred to more professional journals related to endoscopy and cardiovascular disease.

1. Using a fast crosslinking biodegradable adhesive, iCarP could stick to target organs. Whereas, how to prevent the occurrence of organ adhesion? The authors didn't consider this issue in the in vivo experiment.

2. How about the biocompatibility of Chlorella? In addition, Intracardiac injection of Chlorella could bring potential risk of heart bleeding, infection and other possible complications.

3. Thought internal organs and tissues could be illuminated by this iCarP and trocar devices, however, visible light still suffer from low penetration depth, due to the absorption and scattering of tissues. For example, the inside of solid tumor still can not be treated by such a phototherapy strategy.

Reviewer #2 (Remarks to the Author):

Dear author:

In this manuscript, the authors present a biodegradable flexible photonic patch for in vivo phototherapy. The results show that iCarP supports long-term, wide spectrum, continuous/pulsatile, deep penetrating (1.5 cm) illumination on beating animal hearts, without physically damaging the target tissue. iCarP may be a safe and powerful device suitable for internal organs/tissues illumination and associated therapy and diagnosis. The manuscript contains a comprehensive set of content and the experimental results are innovative. Nevertheless, some major issues were raised about the experimental design and results. The mechanisms by which iCarP inhibits cardiomyocyte apoptosis and myocardial fibrosis still remain unclear, and the specific downstream effectors of iCarP has not been elucidated in the study. All these questions need be addressed before its publication in this journal.

Major comments:

1. In Supplementary Fig. 2, the authors showed no interruption in illumination or no significant changes in illumination area when the heart rate > 300 bpm. However, the heart rate in human is less than 300 bpm. Whether will iCarP affect heart rate in human? Especially in pathological conditions, whether will iCarP induce arrhythmia in animals or humans or not?

2. In this manuscript, the continuous transmission of iCarP illumination in the same animal was evaluated by adjusting and controlling different wavelengths of visible light (405, 520, 660 nm). Please explain why other commonly used wavelengths such as 445nm blue light were not chosen?

3. Could the authors exclude that the anti-myocardial fibrosis effect of iCarP is due to the enhancement of survival myocardial cells?

4. What is the mechanism of iCarP inhibiting the proliferation of myocardial fibroblasts? It was suggested to performed in vitro experiments to explore the effect and mechanism of iCarP on the proliferation of cardiac fibroblasts.

5. TUNEL assay showed iCarP inhibits cardiomyocyte apoptosis. Please test the effects of iCarP on the downstream apoptotic molecules in cardiomyocytes.

6. Increased ROS production was involved in light therapy in many cells. In this study, whether will iCarP induce the ROS accumulation in cardiomyocytes or not?

7. In Fig. 5c, the authors showed that iCarP regulates the expression of gap junction protein Cx43 and cTnT. However, the above results do not reflect whether iCarP plays a regulatory role in myocardial contractility. It is necessary to detect the effects of iCarP on myocardial electrophysiology and

myocardial contractility in vitro.

8. In this manuscript, due to the limitation of the duration of anesthesia in mice, the heart illumination should not exceed a maximum of 3 hours and secondary illumination may be used. But how long will the lighting time control cause heart damage?

9. In line 141, "preiously" should be revised into previously.

Reviewer #3 (Remarks to the Author):

The current manuscript presents a flexible, biodegradable photonic device iCarP. The iCarP phototherapy maybe improved survival of cardiac cells and preserved cardiac functions. The results are promising, which is meaningful to the field. But there are several improvements to be made before publication.

1. Why did the author choose 405 nm, 520 nm and 660 nm to irradiate the heart? How do they choose these different wavelengths in the case of different heart diseases?

2. In this study, was the energy used for iCarP lighting the same for rat and dog? If not? How much energy do they use? How are the energy standards of iCarP used for different animals defined?

3. In Figure 1, the author used the model of myocardial infarction induced by intracardiac injection of chlorella, and the ligation model of left anterior descending was used in the following text. Is there any difference between the two models in iCarP irradiation treatment? Why do the authors use these two myocardial infarction models? Please explain.

4. Although the study design seems acceptable, there were concerns I had with the statistical analysis. Was an a-priori sample size/power analysis performed for this study? Given the small sample sizes in each group, I would recommend the use of non-parametric tests, such as a Wilcoxon Rank Sum test for the pairwise tests, or a Kruskal Wallis test for any differences among groups.

5. The author claimed that the lighting depth of iCarP was 1.5 cm. How was this value calculated?

7. I want to know whether the iCarP-slice attached to the heart surface will affect the diastolic and systolic functions of the heart during treatment?

8. Line 113 have an extra "in".

9. The following articles are recommended to be cited by the author. Eur. Radiol 28(5): 2176, 2018; Nat Nanotechnol 16: 455, 2021; Cancer Lett 496: 169, 2021.

10. The images in Figure 3 lack scale bar.

Response to reviewers

Reviewer 1

In this work, the authors presented an iCarP device, which could achieve internal organ illumination. The optical fiber technology upgradation may have some novelty, but most of other findings does not have enough novelty or innovative aspects, for example materials or mechanisms, to be published in this top journal. Therefore, I would suggest this manuscript to be transferred to more professional journals related to endoscopy and cardiovascular disease.

We thank the reviewer for the acknowledgement of our efforts in advancing the optical fiber technology. With the unique air gap structure developed in this study (patent recently issued), “bulb-like” illumination was realized, guiding the light towards the target tissue. The combination of dual reflections and diffraction was introduced as a new strategy to scatter light. Large area, high intensity, wide spectrum, continuous/pulsatile, deeply penetrating illumination was achieved without puncturing the target tissues. Chlorella as a new kind of photosynthesis plant in cardiac applications was reported. Additional experiments were performed to evaluate the risk in patch adhering to surrounding tissues. Please find our responses listed below.

1.1 Using a fast crosslinking biodegradable adhesive, iCarP could stick to target organs. Whereas, how to prevent the occurrence of organ adhesion? The authors didn't consider this issue in the in vivo experiment.

Our response: We thank the reviewer for raising this question. In this study, the fast crosslinking biodegradable adhesive is a novel injectable hydrogel adhesive (CCS@gel) formed by Schiff base reaction between catechol-functionalized chitosan (CCS) and dibenzaldehyde-terminated polyethylene glycol (DB-PEG2000) after mixture¹ (Fang et al. *Carbohydr. Polym.* 2021). The reactive groups of CCS@gel could covalently bind to iCarP soon after contact, and CCS@gel undergoes rapid sol-gel transition as a result of crosslinking between CCS and DB-PEG2000. After complete transformation, the reactive groups are consumed and the gel would not adhere to other tissue.

For example, in iCarP implantation onto heating hearts, the adhesive is applied between iCarP and the epicardium, and the wound is closed after CCS@gel is fully crosslinked and loses adhesiveness. For demonstration, we implanted iCarP onto a rat heart, and examined adhesion between the device and tissue 3 h after surgery. As shown in **Fig. R1** and Supplementary Video 6 (added to the manuscript), CCS@gel attached iCarP to the heart without causing additional adhesion to the chest or other organs.

Figure R1. Evaluation of the risk of iCarP adhesion to surrounding tissues. Neither iCarP adhesion or CCS@gel adhesion to the chest or tissues other than the epicardium were observed. Corresponding video was added as Supplementary Video 6.

Text in the Results section of the manuscript was modified accordingly, at page 20 line 565.

“The biodegradable PMCL substrate remained adhered to the epicardium, exhibiting no detachment (Fig. 5b). ICarP did not adhere to the chest, as the reactive groups of CCS@gel were consumed during crosslinking and covalent bonding with iCarP and epicardium (Supplementary Video 6). The structures of both TOF and PMCL substrate were intact. The firm adherence between the PMCL substrate and the epicardium is an important basis for repeated illumination and extended mechanical support after removing the optical fiber.”

1.2 How about the biocompatibility of Chlorella? In addition, Intracardiac injection of Chlorella could bring potential risk of heart bleeding, infection and

other possible complications.

Our response: We agree with the reviewer that being biocompatibility is an important premise for application of Chlorella. Previous studies have demonstrated the high biocompatibility and low cytotoxicity of Chlorella at concentrations up to 10^6 /mL, and proper surface modification could further improve the biocompatibility of Chlorella^{2,3} (Li et al. *ACS Appl. Mater. Interfaces* 2020; Qiao et al. *Sci. Adv.* 2020).

As the Chlorella suspension is injected into the myocardium with an insulin needle, minor bleeding is inevitable when the needle is pulled out. Compared to PBS injection with an insulin needle, Chlorella suspension injection did not cause extra bleeding. In addition, in animal experiments of this study, including intramyocardial Chlorella injections, rapid hemostasis was achieved by swab pressing. As demonstrated in **Fig. R2** and Supplementary video 4 (added to the manuscript), the minor bleeding from the injection site on a beating rat heart was stopped within a few seconds. In addition, CCS@gel seals the wound, and the implanted iCarP keeps a pressure on the wound, therefore also having hemostatic effects.

The Chlorella used in this work were cultured in sterile conditions. No bacterial activities or colonies were observed in extended cell culture with Chlorella, which shows the sterility of the Chlorella suspension. In our rat studies, no festering was observed on Chlorella injected hearts, showing a low risk of infection. In addition, we evaluated the expression levels of inflammation markers, IL-6 and TNF- α in the treated hearts. As shown in **Fig. R2** and Supplementary Fig. 8 (added to the manuscript), IL-6 and TNF- α expression in and near the Chlorella injection sites are not significantly higher compared to expression in the interventricular septum and RV myocardium.

In summary, the animal studies showed that Chlorella injection is a safe procedure and does not pose significant bleeding, infection or inflammation risks.

Figure R2. Evaluation of the potential risks of bleeding and inflammation caused by Chlorella injection. **a**, the minor bleeding caused by the insulin needle was stopped by swab pressing within seconds. Corresponding video was added as Supplementary video 4. **b,c**, Representative immunofluorescent staining images of IL-6 (**b**) and TNF- α (**c**) in rat hearts 3 d after Chlorella injection in LV myocardium, scale bar = 2 mm. Inset: magnified images, scale bar = 50 μ m.

Text in the Materials and methods and Results section of the manuscript was modified accordingly, at page 9 line 243, and at page 20 line 556.

“10 min after LAD ligation, Chlorella (100 μ L, 2×10^7 /mL) suspension was injected into infarcted LV of rats in iCarP groups, followed by swab pressing for hemostasis, and iCarP adhesion via CCS@gel on the epicardium of the infarct area. Subsequent to iCarP implantation, chest cavity, muscles and skin were sutured with 3-0 silk sutures.”

“The entire infarcted rat myocardium turned from pale white to light green after intramyocardial Chlorella injection, showing that the injected Chlorella distributed throughout the infarct (data not shown). No major bleeding occurred during or after Chlorella injection as shown in Supplementary Fig. 8 and Supplementary Video 4. Limited by the anesthesia time, Chlorella was continuously illuminated for 3 h after injection in closed chest state (continuous illumination could theoretically be extended if allowed).”

1.3 Thought internal organs and tissues could be illuminated by this iCarP and trocar devices, however, visible light still suffer from low penetration depth, due to the absorption and scattering of tissues. For example, the inside of solid tumor

still can not be treated by such a phototherapy strategy.

Our response: We agree with the reviewer’s concern. In various phototherapies, the penetration depth is limited mainly due to (1) light shielding by skin, bones and other soft tissues between the light source and target tissue, and (2) light absorption and scattering by tissues. As iCarP could be directly implanted onto the target tissue, as demonstrated in our study with iCarP adhesion to heart, muscle, liver, and subcutaneous implantation close to tumors, optical energy loss attributed to light shielding could be avoided. Such strategy has been adopted in other studies, including the work by Yamagishi, et al., in which they implanted a wirelessly powered LED light source subcutaneously for tumor phototherapy⁴ (Yamagishi et al. *Nat. Biomed. Eng.* 2018).

On the other hand, limited visible light penetration due to tissue absorption and scattering is inevitable. Increasing the optical power could increase penetration depth. The optical fiber in iCarP supports transmittance of high-power light. In addition, iCarP supports transmittance of light with wavelengths between 400 nm and 2000 nm, covering the NIR-I and NIR-II window. Light in the NIR-I and NIR-II windows has significantly greater tissue penetration depth⁵ (Lee et al. *Nat. Rev. Mater.* 2020). Theoretically, with matched upconversion materials in the target tissue, iCarP can be used in phototherapy of light with short wavelength.

Figure R3. a, Construction of tissue-adhesive optoelectronics composed of an NFC-based LED chip sandwiched between PDA–PDMS and pristine PDMS nanosheets, and implantation of the device on the target lesion. **b**, Image of the implantation of the device onto the inner surface of the dorsal skin of a mouse. From *Nat Biomed Eng.* 2018; 3: 27–36⁴.

Text in the Discussion section of the manuscript was modified accordingly, at page 24 line 692.

“The iCarP design principle is theoretically compatible with optical fibers and

photonics patches that deliver lights with wavelength < 400 nm. For light in the tissue penetration window, the optical fiber iCarP could transmit light with significantly higher intensity and spatial precision, as there is no energy dissipation in tissue between the target and light source given that iCarP is directly attached to the target tissue.”

Reviewer 2

In this manuscript, the authors present a biodegradable flexible photonic patch for in vivo phototherapy. The results show that iCarP supports long-term, wide spectrum, continuous/pulsatile, deep penetrating (1.5 cm) illumination on beating animal hearts, without physically damaging the target tissue. ICarP may be a safe and powerful device suitable for internal organs/tissues illumination and associated therapy and diagnosis. The manuscript contains a comprehensive set of content and the experimental results are innovative. Nevertheless, some major issues were raised about the experimental design and results. The mechanisms by which iCarP inhibits cardiomyocyte apoptosis and myocardial fibrosis still remain unclear, and the specific downstream effectors of iCarP has not been elucidated in the study. All these questions need be addressed before its publication in this journal.

We thank the reviewer for the acknowledgement on the novelty of our study. To answer the reviewer's questions about the mechanism of iCarP illumination inhibiting apoptosis and fibrosis. Immunofluorescent staining and RNA-seq were employed to reveal the changes in expression levels of marker proteins for apoptosis, and marker genes for apoptosis and fibrosis, before and after iCarP triggered photosynthesis in MI rats. We also performed in vitro studies to evaluate the influence of illumination on cardiomyocyte electrophysiology and activity, fibroblast proliferation, and ROS accumulation in cells. Corresponding results are supplemented below.

2.1 In Supplementary Fig. 2, the authors showed no interruption in illumination or no significant changes in illumination area when the heart rate > 300 bpm. However, the heart rate in human is less than 300 bpm. Whether will iCarP affect heart rate in human? Especially in pathological conditions, whether will iCarP induce arrhythmia in animals or humans or not?

Our response: We agree with the reviewer's concern about the differences in the heart rates between rats and human, and the risk of iCarP induced arrhythmia. To answer this question, we studied the influence of iCarP illumination on canine (before and after MI) heart rate. Electrocardiography was used to record the heart rate of the dog before and

after iCarP illumination. In addition, we integrated a U-shaped optical fiber into iCarP to turn iCarP into a waveguiding motion sensor, and the sensor was used to confirm the electrocardiography results. As the heart beats, the U-shaped optical fiber deforms, resulting in the cyclic intensity fluctuation of the guided light returned to the receiver.

ECG and U-iCarP were used simultaneously to record the heart rate of the Beagle we used. It had a baseline heart rate of 96 bpm, as shown in **Fig. R4** (added to the manuscript as Supplementary Fig. 14). After implanting U-iCarP and illumination with 660 nm light, the heart rate remained unchanged at 96 bpm. Interestingly, U-iCarP also captured the motion signal of breaths (the larger peaks on **Fig. R4i-I** marked by blue arrows). Then, we ligated LAD and LCx to create the infarct model, which was confirmed as the ischemic area became pale. The heart rate slightly increased to 102 bpm. A second round of iCarP illumination was performed to evaluate the risk of inducing arrhythmia. During the 5 min recording, no arrhythmia was observed by ECG or U-iCarP, which is consistent with the finding in the rat study. The heart rate was remained unchanged at 102 bpm. It is worth noting that we transmitted red light in the tapered optical fiber, and green light in the U-shaped optical fiber, therefore the canine experiment demonstrated that the risk of iCarP illumination induced arrhythmia is negligible, at least for red and green lights.

Results in the literature also support our claim. Nussinovitch and Gepstein used adeno-associated virus 9 (AAV9) to express the Channelrhodopsin-2 (ChR2) transgene at one or more ventricular sites in rats⁶ (Nussinovitch et al. *Nat. Biotechnol.* 2015). The authors achieved optogenetic pacing of the hearts at different beating frequencies with blue-light illumination both in vivo and in isolated perfused hearts. In contrast, no optogenetic pacing occurred during illumination to remote myocardial areas in the same hearts or in animals injected with control AAV virus⁶.

Figure R4. Effects of iCarP on heart rate of a dog. **a**, Preoperative preparation. **b**, Thoracotomy & iCarP implantation. **c**, Ligation of LAD and LCx. **d**, iCarP implantation & illumination. **e-l**, Dog heart rate measured by electrocardiogram (e-h) and U-iCarP (i-l) at different stages of the surgery, including before illumination (before ligation) (e,i), during illumination (before ligation) (f,j), before illumination (after ligation) (g,k) and during illumination (after ligation) (h,l). **m**, Structural diagram of U-iCarP. **n**, Digital image of U-iCarP. **o**, Application of U-iCarP on the dog heart.

Text in the Results section of the manuscript was modified accordingly, at page 23 line 648.

“The attached iCarP moved along with the epicardium, showing no detachment or sliding, despite the significantly greater amplitude of heart contraction. The illumination stayed stable and evenly distributed in the patch covered area during the entire experiment (Fig. 6d, iv). The influence of iCarP illumination on heart rate of healthy and MI dogs was evaluated. As shown in Supplementary Fig. 14 and Supplementary Video 7, iCarP illumination did not affect heart rate of the dog, or cause arrhythmia. These results showed the adaptability and safety of iCarP with minimally invasive implantation surgeries on other internal organs and tissues.”

2.2 In this manuscript, the continuous transmission of iCarP illumination in the same animal was evaluated by adjusting and controlling different wavelengths of

visible light (405, 520, 660 nm). Please explain why other commonly used wavelengths such as 445 nm blue light were not chosen?

Our response: We thank the reviewer for raising the question about the compatibility of iCarP with other commonly used wavelengths. To answer this question, we connected iCarP to a 445 nm light source. As shown in **Fig. R5** (added to the manuscript as Supplementary Fig. 5), 445 nm light can be transmitted and scattered with iCarP. As previously described in our manuscript (**Fig. 2**), light with wavelength of 405, 520 and 660 nm were chosen as representative wavelengths of light to demonstrate the compatibility of iCarP with different wavelengths. Furthermore, light with wavelength of 473 nm and 808 nm were also used in PDT and PPT for tumor (Supplementary Fig. 6, **Fig. 3**). The tests above cover a wide range of wavelengths in which commonly used lights fall in. According to the transmission spectrum of the optical fiber and PMC, iCarP is theoretically compatible with visible and near-infrared light, which should meet the wavelength requirements of mainstream phototherapies.

Figure R5. iCarP guided light with wavelength of 445 nm onto an ex vivo porcine heart. Scale bar = 5 mm.

Text in the Results section of the manuscript was modified accordingly, at page 15 line 424.

“The divergence angles of shorter wavelength lasers were smaller than 660 nm laser, which is believed due to the smaller refractive indexes of iCarP components including the PMCL patch corresponding to the blue and green laser, compared to the refractive index of the red laser. In addition, the compatibility of iCarP with lasers of 445 nm

(Supplementary Fig. 5), 473 nm (Supplementary Fig. 6), and 808 nm (Fig. 4) wavelengths was also demonstrated. The wavelengths used above are highly representative in current phototherapies, which showed that iCarP illumination meet the wavelength requirements in photo-responsive drug delivery, photodynamic therapy, photothermal therapy, optogenetics, etc.”

2.3 Could the authors exclude that the anti-myocardial fibrosis effect of iCarP is due to the enhancement of survival myocardial cells?

Our response: We thank the reviewer for raising this question about the therapeutic effects of iCarP facilitated in situ photosynthesis. We apologize for the confusion in explaining the anti-fibrosis mechanism. We think the observed anti-fibrosis effect was one of the outcomes of enhanced survival of myocardial cells attributed to in situ oxygen production.

Figure R6. Biphasic nature of cardiac repair after myocardial infarction (MI), from *Circ Res.* 2016; 119: 91–112⁷.

As discussed in the review by Prabhu and Frangogiannis⁷, deaths of cardiac cells induce inflammation responses, which subsequently activates fibroblasts (Fig. R6). Efforts have been made to decrease post-MI fibrosis by rescuing cardiac cells, particularly cardiomyocytes⁸ (Wu et al. *Cell Death Dis.* 2020). Fig. 5 in our manuscript

demonstrated iCarP facilitated in situ oxygen production significantly reduced apoptosis cardiac cells, and inhibited fibrosis of infarcted LV myocardium.

To further illustrate the anti-apoptosis and anti-fibrosis effects of the iCarP facilitated phototherapy, we ran an RNA-seq and analyzed the RNA expression of Sham, MI, and iCarP+/Light+ treated rat hearts as shown in **Fig. R7** (added to the manuscript as Supplementary Fig. 10). Principal component analysis (PCA) demonstrated that iCarP triggered in situ photosynthesis (iCarP+/Light+) significantly altered the transcriptome compared with the MI group as shown in **Fig. R7a**. The volcano graph visualized differentially expressed genes (DEGs) between iCarP+/Light+ group and MI group (**Fig. R7b**) ($\text{Log}_2(\text{fold change}) > 1$ or < -1 , adjusted p value < 0.05). To further explore the therapeutic function of iCarP+/Light+ treatment, the Sham and MI group was first compared to select DEGs. Among these DEGs, the genes whose expression levels in iCarP+/Light+ group recovered from MI (iCarP+/Light+ vs MI and Sham vs MI in the same trend), defined as iCarP effective genes. Clustering analysis of the iCarP effective genes was shown in **Fig. R7c**. The iCarP+/Light+ group clustered with the Sham group and separated with the MI group. Gene Ontology (GO) Biological Process (BP) analysis revealed that up-regulated genes in the Sham and iCarP+/Light+ group compared to the MI group were involved in negative regulation of NF-kappaB transcription factor activity, negative regulation of I-kappaB kinase/NF-kappaB signaling, negative regulation of apoptotic process and negative regulation of inflammatory response, indicating that iCarP+/Light+ could protect the LV myocardium against MI by reducing apoptosis, which is consistent with the histological results in **Fig. 5**. The down-regulated genes were involved in neutrophil activation, T cell activation, adaptive immune response and natural killer cell activation, revealing that in situ photosynthesis could reduce inflammation (**Fig. R7d**). The expression levels of representative genes, Bax, Htra2 and Myc for apoptosis; Tgfb1, Tgfb2 and Col5a3 for fibrosis, are shown in **Fig. R7e**. These results support the claims in the manuscript that iCarP triggered in situ photosynthesis could reduce apoptosis and fibrosis to protect infarcted myocardium.

Figure R7. RNA-seq analysis of cardiac tissues 1 day after surgery. **a**, Principal component analysis (PCA) of transcriptomes of LV myocardium from MI, iCarP+/Light+ and Sham groups (n = 4). **b**, Volcano plot of all expressed genes from the iCarP+/Light+ and MI group (n = 4 per group). The vertical dashed lines represent the border of gene expressions in the iCarP+/Light+ group greater than 1-fold change vs the MI group. Dots above the horizontal dashed line have adjusted $p < 0.05$. **c**, Clustering analysis of the Sham, MI, and iCarP+/Light+ groups using apoptosis-related genes (Increased group, up-regulated genes in Sham and iCarP+/Light+ groups compared to in MI group; Decreased group, down-regulated genes in Sham and iCarP+/Light+ groups compared to in MI group; n = 4 per group). **d**, Gene Ontology (GO) Biological Process (BP) analyses of iCarP effective genes. **e**, Expression levels of Bax, Htra2, Myc, Tgfb1, Tgfb2 and Col5a3 genes in Sham, MI, and iCarP+/Light+ groups (n = 4 per group). Statistical significance was calculated using unpaired t-test and data are presented as means \pm SEM. * $p < 0.05$, ** $p < 0.01$, *** $p < 0.0001$ vs MI group.

Text in the Results section of the manuscript was modified accordingly, at page 21 line 581, at page 21 line 590.

"In addition, the expressions of pro-apoptotic Bax and Caspase-3 were lowest in

iCarP+/Light+ group, and the *Bax* and *Caspase-3* levels were significantly lower in *iCarP+/Light-* group compared to them of MI group. RNA-seq showed that the expression levels of representative pro-apoptosis genes in LV myocardium, including *Bax*, *Htra2* and *Myc* were elevated by MI compared to baseline levels in Sham rats, and were decreased by in situ photosynthesis in *iCarP+/Light+* group.”

“Myocardial fibrosis detected by Masson’s trichrome staining 28 days post-surgery (Fig. 5e) exhibited a trend same to apoptosis results. which is expected as fibrosis is considered a result of cardiac damage⁴⁷. Compared to MI group (25.6±2.8%), *iCarP+/Light-* treatment significantly reduced cardiac fibrosis (22.1±2.0%), while *iCarP+/Light+* group had the lowest fibrosis level (20.1±2.5%). Muscle tissue can be found stained in red in infarcted left ventricle from *iCarP+/Light+* treated hearts, whereas the entire infarcted LV was fibrotic in hearts from MI group (Fig. 5c). The expression levels of representative fibrosis genes, including *Tgfb1* and *Tgfb2* for TGF- β signaling pathway and *Col5a3* for collagen fibril organization were elevated in MI group compared to Sham group, and decreased in *iCarP+/Light+* group, supporting the effects of *iCarP* triggered in situ photosynthesis on fibrosis in reducing fibrosis (Supplementary Fig. 10).”

2.4 What is the mechanism of iCarP inhibiting the proliferation of myocardial fibroblasts? It was suggested to performed in vitro experiments to explore the effect and mechanism of iCarP on the proliferation of cardiac fibroblasts.

Our response: We thank the reviewer for raising this question and apologize for the confusion. We did not claim that *iCarP* facilitated in situ photosynthesis inhibited the proliferation of myocardial fibroblasts, although it may be true in our design. We intended to save cardiomyocytes from the ischemic event by *iCarP* treatment, thus attenuate the pathological remodeling of LV. Reduced LV fibrosis, including the changes in the expression levels of fibrosis associated genes (Fig. R7), is one of the phenomena of attenuated remodeling. As discussed in the original manuscript and in

the answer for the last question, we incline to associate the reduced fibrosis to inhibited apoptosis. In addition to in situ oxygen production, mechanical support from iCarP also contributed to the reduction in fibrosis, as shown in **Fig. 5** of the manuscript, and by previous studies in the literature^{9,10} (Lin et al. *Nat. Biomed. Eng.* 2019; Montgomery et al. *Nat. Mater.* 2017).

To evaluate the influence of iCarP illumination and in situ photosynthesis on fibroblast proliferation, we carried out an in vitro cell culture experiment. The results showed that none of Chlorella, iCarP illumination, or their combination significantly changed the viability or proliferation rate of primary cardiac fibroblasts fibroblasts (**Fig. R8**, added to the manuscript as Supplementary Fig. 12).

Figure R8. Effect of Chlorella and light on cardiac fibroblasts proliferation.

Text in the Results section of the manuscript was modified accordingly, at page 21 line 590.

“Myocardial fibrosis detected by Masson’s trichrome staining 28 days post-surgery (**Fig. 5e**) exhibited a trend same to apoptosis results, **which is expected as fibrosis is considered a result of cardiac damage⁴⁷**. Compared to MI group ($25.6\pm 2.8\%$), *iCarP+/Light-* treatment significantly reduced cardiac fibrosis ($22.1\pm 2.0\%$), while *iCarP+/Light+* group had the lowest fibrosis level ($20.1\pm 2.5\%$). Muscle tissue can be found stained in red in infarcted left ventricle from *iCarP+/Light+* treated hearts, whereas the entire infarcted LV was fibrotic in hearts from MI group (**Fig. 5c**). **The expression levels of representative fibrosis genes, including *Tgfb1* and *Tgfb2* for TGF- β signaling pathway and *Col5a3* for collagen fibril organization were elevated in MI**

group compared to Sham group, and decreased in iCarP+/Light+ group, supporting the effects of iCarP triggered in situ photosynthesis on fibrosis in reducing fibrosis (Supplementary Fig. 10). In vitro cell culture showed that none of Chlorella, iCarP illumination, or their combination significantly changed the viability or proliferation rate of primary fibroblasts (Supplementary Fig. 12), again indicating inhibiting cardiac cell apoptosis may be the primary reason for observed reduction in fibrosis.”

2.5 TUNEL assay showed iCarP inhibits cardiomyocyte apoptosis. Please test the effects of iCarP on the downstream apoptotic molecules in cardiomyocytes.

Our response: We agree with the reviewer that inhibition of cardiomyocyte apoptosis is one of the key outcomes of iCarP facilitated in situ oxygen production, thus worth more investigation. In addition to RNA-seq resented above, we also did immunofluorescent staining of apoptosis markers, Bcl-2, Bax and Caspase-3. Among the three markers, Bcl-2 inhibits apoptosis, Bax is a pro-apoptotic regulator, and Caspase-3 is a frequently activated death protease in apoptosis. As shown in **Fig. R9**, the expression of Bcl-2 in iCarP+/Light- group was significantly higher than that in the MI group, and the expression in iCarP+/Light+ group was the highest among the three groups. In addition, the expressions of Bax and Caspase-3 were lowest in iCarP+/Light+ group, and the Bax and Caspase-3 levels were significantly lower in iCarP+/Light- group compared to them of MI group. These results are consistent with TUNEL and RNA-seq results, which confirmed that mechanical support and iCarP facilitated in situ oxygen production inhibited apoptosis of cardiac cells.

Figure R9. Expression of apoptosis markers in myocardium. Representative immunofluorescence staining images of cTnT /Bcl-2 (a), cTnT /Bax (b) and cTnT /Caspase-3 (c) in left ventricular 1 d after MI (nuclei: blue, cTnT: red, Bcl-2/Bax/Caspase-3: green). Scale bars = 200 μ m. Quantitative analysis of expression of Bcl-2 (d), Bax (e) and Caspase-3 (f).

Text in the Results section of the manuscript was modified accordingly, at page 20 line 576.

“Between the two iCarP treated groups, the iCarP+/Light+ group showed a lower percentage of apoptotic cells compared to iCarP+/Light- group (Fig. 5d), attributed to oxygen produced by light triggered photosynthesis. Immunofluorescent staining of apoptosis related molecules Bcl-2, Bax, and Caspase-3 exhibited consistent results. As shown in Supplementary Fig. 9, expression level of anti-apoptotic Bcl-2 was reduced by MI compared to that in Sham group, and was highest in iCarP+/Light+ group. In addition, the expression levels of pro-apoptotic Bax and Caspase-3 were lowest in iCarP+/Light+ group, significantly lower compared to them of MI group. RNA-seq

showed that the expression levels of representative pro-apoptosis genes in LV myocardium, including *Bax*, *Htra2* and *Myc* were elevated by MI compared to baseline levels in Sham rats, and were decreased by in situ photosynthesis in *iCarP+/Light+* group.”

2.6 Increased ROS production was involved in light therapy in many cells. In this study, whether will iCarP induce the ROS accumulation in cardiomyocytes or not?

Our response: We thank the reviewer for raising this question about the risk of iCarP illumination inducing excessive ROS. To evaluate such risk, we illuminated H9C2 cells in vitro with and without Chlorella, and measured the ROS levels. Results demonstrated that iCarP illumination-triggered oxygen production did not induce intracellular ROS accumulation in cardiomyocytes under the experimental conditions in this study (wavelength of 660 nm, power of 55 mW, 3 h), Chlorella concentration of 10^6 /mL) (Fig. R10, added to manuscript as Supplementary Fig. 11).

Figure R10. Effect of Chlorella and light on ROS accumulation in cardiomyocytes, which was stained by DCFH-DA.

Text in the Results section of the manuscript was modified accordingly, at page 21 line 584.

“RNA-seq showed that the expression levels of representative pro-apoptosis genes in LV myocardium, including *Bax*, *Htra2* and *Myc* were elevated by MI compared to baseline levels in Sham rats, and were decreased by in situ photosynthesis in *iCarP+/Light+* group. *In vitro cell culture showed that illumination (660 nm, 55 mW,*

3 h) with or without Chlorella did not induce intracellular ROS accumulation in cardiomyocytes, as shown in Supplementary Fig. 11. Consecutive illumination for 12 h did not decrease the viability of cardiomyocytes, indicating the safety of extended iCarP illumination in vivo.”

2.7 In Fig. 5c, the authors showed that iCarP regulates the expression of gap junction protein Cx43 and cTnT. However, the above results do not reflect whether iCarP plays a regulatory role in myocardial contractility. It is necessary to detect the effects of iCarP on myocardial electrophysiology and myocardial contractility in vitro.

Our response: We appreciate the reviewer for raising this question. To detect the effects of iCarP on myocardial electrophysiology and myocardial contractility in vitro, E-Plate Cardio 96 (Agilent Technologies Inc, USA) and xCELLigence RTCA Cardiosystem system (Agilent Technologies Inc, USA) were used to monitor electrophysiology and contractility of primary cardiomyocytes before, during, and after iCarP illumination. In the experiment set up, microelectrodes (MEs) and interdigitated electrodes (IDEs) were used to record the extracellular action potential (EAP), and mechanical beating (MB) induced by excitation frequency of IDEs, respectively, as previously described¹¹ (Wei et al. *ACS Sens.* 2021). Results showed that iCarP illumination, Chlorella, and the combination of both did not significantly affect amplitude and cell index of the cardiomyocytes (**Fig. R11**, added to the manuscript as Supplementary Fig. 13). Compared to Chlorella+/light-, Chlorella+/light+ increased firing rate, which is believed to be attributed to the increased energy supply from Chlorella photosynthesis-induced oxygen production.

Figure R11. Effects of Chlorella and light on myocardial electrophysiology and myocardial contractility in vitro. **a**, Schematic diagram of the sensor for simultaneous detection of primary cardiomyocyte extracellular action potential and mechanical beating. **b**, Electrode map of microelectrodes and interdigitated electrodes. **c**, Schematic diagram of the experiment and definition of signal characteristic parameters. EE1 and EE2 were defined to label the peak, valley of extracellular action potential. MB1 and MB2 were defined to label the peak, valley of mechanical beating. **d-f**, Effect of Chlorella and light on EAP amplitude (d), MB amplitude (e) and firing rate (f) in primary cardiomyocytes.

Text in the Results section of the manuscript was modified accordingly, at page 22 line 615.

“The expression level and organization of cTnT and Cx43 in infarcted myocardium improved significantly after being treated by iCarP+/Light+ treatment (Fig. 5c), indicating a healthier myocardium supported by local oxygen supply. The effects of iCarP on myocardial electrophysiology and myocardial contractility in vitro were evaluated by multifunctional hybrid integrated cardiomyocyte biosensor⁴⁸, as shown in Supplementary Fig. 13. Results showed that illumination and Chlorella did not affect cardiomyocyte contraction and relaxation, but the firing rate was higher with Chlorella,

which is believed to be a result of increased energy supply from photosynthesis.”

2.8 In this manuscript, due to the limitation of the duration of anesthesia in mice, the heart illumination should not exceed a maximum of 3 hours and secondary illumination may be used. But how long will the lighting time control cause heart damage?

Our response: We thank the reviewer for raising this question. According to the American National Standard for Safe Use of Optical Fiber Communication Systems Utilizing Laser Diode and LED Sources (Laser Institute of America, Orlando, FL, 2012), at the powers used in our study (except the tumor ablation experiments in which cancer cell damage is desired), iCarP illumination theoretically would not cause tissue injury after 10 h and longer consecutive usage. For example, the powers used in the MI treatment experiments were > 50% lower compared to the power used in the Standard for calculation of the duration thresholds, thus iCarP illumination should be safe even after consecutive use for days. To verify this analysis, we performed a 12 h illumination experiment. As shown in **Fig. R12**, extended illumination did not decrease the viability of cardiomyocytes.

Figure R12. Effect of extended illumination on cardiomyocyte viability.

Text in the Results section of the manuscript was modified accordingly, at page 21 line 586.

“In vitro cell culture showed that illumination (660 nm, 55 mW, 3 h) with or without Chlorella did not induce intracellular ROS accumulation in cardiomyocytes, as shown in Supplementary Fig. 11. Consecutive illumination for 12 h did not decrease the viability of cardiomyocytes, indicating the safety of extended iCarP illumination in vivo.”

2.9 In line 141, “preiously” should be revised into previously.

Our response: We thank the reviewer for pointing out the typo. Correction has been made as

“Then, end diol groups were acrylated with excess acryloyl chloride and triethylamine. CCS@gel was prepared as previously described (Supplementary Fig. 2)⁴⁰”.

Reviewer 3

The current manuscript presents a flexible, biodegradable photonic device iCarP. The iCarP phototherapy maybe improved survival of cardiac cells and preserved cardiac functions. The results are promising, which is meaningful to the field. But there are several improvements to be made before publication.

We thank the reviewer for the acknowledgement of our study. To answer the reviewer's questions, we performed extra experiments, including 405 nm illumination with iCarP, and finite element analysis of the mechanical effect of a cardiac patch. Our responses are listed below.

3.1 Why did the author choose 405 nm, 520 nm and 660 nm to irradiate the heart? How do they choose these different wavelengths in the case of different heart diseases?

Our response: We thank the reviewer for raising this insightful question. Light with wavelength of 405, 520 and 660 nm were as representatives to demonstrate the compatibility of iCarP with light from a wide range of spectrum. In therapeutic applications, the wavelength of delivered light were chosen according to the sensitive wavelengths of the photo-active agents. In iCarP illumination triggered photosynthesis, 660 nm was chosen as it is within the second absorption peak of chlorophyll. Furthermore, 473 nm and 808 nm lasers were chosen to match with the photosensitizer and photothermal agent used in the tumor PDT and PPT, respectively. Theoretically, iCarP can be applied to deliver light for most phototherapies, as the working wavelengths of commonly used photosensitive agents fall in the 400-2000 nm range⁶, which is supported by iCarP.

Text in the Results section of the manuscript was modified accordingly, at page 15 line 424.

“The divergence angles of shorter wavelength lasers were smaller than 660 nm laser, which is believed due to the smaller refractive indexes of iCarP components including the PMCL patch corresponding to the blue and green laser, compared to the refractive index of the red laser. In addition, the compatibility of iCarP with lasers of 445 nm

(Supplementary Fig. 5), 473 nm (Supplementary Fig. 6), and 808 nm (Fig. 4) wavelengths was also demonstrated. The wavelengths used above are highly representative in current phototherapies, which showed that iCarP illumination meet the wavelength requirements in photo-responsive drug delivery, photodynamic therapy, photothermal therapy, optogenetics, etc.”

3.2 In this study, was the energy used for iCarP lighting the same for rat and dog? If not? How much energy do they use? How are the energy standards of iCarP used for different animals defined?

Our response: We thank the reviewer for raising this question. The energy was comprehensively considered in view of the tolerance of animals and the energy required for corresponding experiments. One of the aims of experiments on dogs and ex vivo pig hearts was to demonstrate the light penetration depth in large animals' hearts and potentially in human hearts, so the power of illumination needs to be higher. In rat experiment, the power of light needs to be high enough to ensure illumination of 2-3 mm thick myocardium. Tumors in the mouse experiments were thicker than rat myocardium, the power of light need to be high enough to induce photosensitizer to produce free radicals and induce photothermal materials to produce sufficient heat, respectively. The above powers are summarized in **Table R1**.

Table R1. The energy used for iCarP illumination in animals in this study

	Dog	Porcine heart	Rat	Mouse (PTT)	Mouse (PDT)
Power (mW)	75	75	55	300	100

Description of the powers used in different animals were included into the:

The power of laser used in canine experiments was described in Materials and methods in the main text, at page 11 line 292.

*“...then the scissors were substituted by iCarP for attachment. **iCarP illumination (75 mW) was tested, all procedure was observed through endoscope.**”*

The power of laser used in porcine heart experiments was described in Materials

and methods in the main text, at page 8 line 214.

“...iCarP was completely covered underneath, turn on the light (75 mW) and observe the penetration depth.”

The power of laser used in the rat heart experiments was added in Materials and methods of Manuscript, at page 9 line 246.

“Rat hearts in iCarP+/Light+ were illuminated continuously for 3 h (55 mW), no illumination was performed in other groups.”

The power used in Mouse (PTT) was added to Materials and methods of Manuscript, at page 8 line 226.

“Solid-state laser with wavelength of 808 nm (DE51543, Changchun New Industries Optoelectronics Tech, China) were used as the light source (300 mW).”

The power used in Mouse (PDT) was described in Supplementary Note 6 of Supporting information, at page 4 line 105.

“Solid-state laser with wavelengths of 473 nm (CF60070, Changchun New Industries Optoelectronics Tech, China) were used as light sources for FEOF and iCarP (100 mW).”

3.3 In Figure 1, the author used the model of myocardial infarction induced by intracardiac injection of chlorella, and the ligation model of left anterior descending was used in the following text. Is there any difference between the two models in iCarP irradiation treatment? Why do the authors use these two myocardial infarction models? Please explain.

Our response: We thank the reviewer for raising this question. All rat MI models in this work were created by LAD ligation. Injection of Chlorella suspension was used for in situ photosynthesis against the hypoxic environment after acute MI^{2,3}. The following figure is the flow chart of rat experiments, 10 mins later LAD ligation, Chlorella was injected and iCarP was implanted

Figure R13. Timeline of the animal study. (from Fig. 5a in Manuscript).

3.4 Although the study design seems acceptable, there were concerns I had with the statistical analysis. Was an a-priori sample size/power analysis performed for this study? Given the small sample sizes in each group, I would recommend the use of non-parametric tests, such as a Wilcoxon Rank Sum test for the pairwise tests, or a Kruskal Wallis test for any differences among groups.

Answer: We agree with the reviewer that proper statistical analysis is critical. Before animal experiments, GPower was used to evaluate the sample size. Results showed that at least 4 samples per group were needed to reveal the differences. After the formal experiments, the data were analyzed, and verified that 4 per group was sufficient. At the same time, we also performed the normal distribution test and the homogeneity of variance test, the results met the requirements of One-way or ANOVA.

Text in the Statistics section of the manuscript was modified accordingly, at page 11 line 297.

“Statistical analysis was performed using SPSS, GPower and Prism software. Results are shown as mean ± standard deviation. T-tests were used to compare results from two groups. One-way analysis of variance (ANOVA) was used for multiple comparisons unless stated otherwise. p value of < 0.05 was considered statistically significant.”

3.5 The author claimed that the lighting depth of iCarP was 1.5 cm. How was this value calculated?

Our response: We thank the reviewer for raising this technical question. We apologize that such information was not clearly provided. As shown in Fig. R14, we placed myocardial blocks with thickness of 1.5 cm on different devices, and the light from

iCarP penetrating the myocardial blocks was visible to naked eye. Given this result, we stated that light from iCarP penetrated 1.5 cm myocardium.

Figure R14. Illumination scope and depth of iCarP. a-c, Panorama (i) and magnified images (ii) of ex vivo porcine hearts illuminated by iCarP. (i) scale bars = 2.5 cm; (ii) scale bar = 1 cm. White arrows indicate the optical fibers. Top view (iii) and front view (iv) of 1.5 cm thick heart tissue illuminated by iCarP (the devices were completely covered by the heart tissue; their locations were indicated by white circles), scale bar = 1 cm.

Text in the Materials and methods section of the manuscript was modified accordingly, at page 8 line 213.

“The iCarP was adhered onto the dissected porcine heart ex vivo to observe the illumination depth at a view of longitudinal section. The porcine heart was cut into cuboids of 1.5 cm thicknesses where iCarP was completely covered underneath, turn on the light and observe the penetration depth.”

3.6 I want to know whether the iCarP-slice attached to the heart surface will affect the diastolic and systolic functions of the heart during treatment?

Our response: We thank the reviewer for raising this question. It has been proven that cardiac patches with appropriate mechanical properties can relieve the stress in the ventricular wall, thereby preserving the cardiac function and attenuating pathological cardiac remodeling^{9,10} (Lin et al. *Nat. Biomed. Eng.* 2019; Montgomery et al. *Nat. Mater.* 2017). The same effect was observed in our study, as rats of iCarP+/Light- group showed a trend of increase in LVEF compared to those of MI group (**Fig. R15a-c**). In addition, we ran a finite element analysis on porcine hearts, which showed that the

infarct has an elevated strain compared to healthy control, and patch implantation could reduce the strain back to the normal range (**Figure R15d-f**).

Figure R15. ICarP-induced improvement of myocardial functions 4 weeks after implantation. Representative M-mode echocardiography images of MI (a) and iCarP+/Light+ (b) rat. c, Quantitative analysis of ejection fraction based on day 28 echocardiography. Finite element analysis of MI (d) and patched (e) porcine hearts. f, Strain change in a diastole-systole cycle.

Text in the Results section of the manuscript was modified accordingly, at page 21 line 604.

“Left ventricle ejection fraction (LVEF) significantly decreased in MI group compared to Sham, indicating cardiac dysfunction after myocardial infarction. Although there was an increasing trend, LVEF of iCarP+/Light- group (with theoretical mechanical support) was not significantly higher compared to LVEF of MI group, despite the short-term cell protection effect shown in TUNEL staining.”

3.7 Line 113 have an extra "in".

Our response: We thank the reviewer for correcting the typo. Correction has been made as

“Stability of iCarP illumination on mechanically challenging, cyclically contracting myocardium was studied **in situ** photosynthesis for MI treatment (**Fig. 1c,d**).”

3.8 The following articles are recommended to be cited by the author. Eur. Radiol 28(5): 2176, 2018; Nat Nanotechnol 16: 455, 2021; Cancer Lett 496: 169, 2021.

Our response: We thank the reviewer for providing more background knowledge. The articles have been cited in the Introduction.

1. **Eur. Radiol 28(5): 2176, 2018; Cancer Lett 496: 169, 2021** were cited in “*Illuminating internal organs and tissues has widely supported optical diagnosis, laser surgery, and other light-activated therapies by creating unique lightened microenvironments on targets¹⁻⁷.*”

2. **Nat Nanotechnol 16: 455, 2021** was cited in “*integrated with endoscopes, or guided by magnetic resonance imaging, optical fiber devices could be minimally invasively placed at desired locations, for targeted illumination in applications including laser-induced phototherapy, optogenetics, temperature and pressure sensing, etc²⁷⁻²⁹.*”

3.9 The images in Figure 3 lack scale bar.

Our response: We thank the reviewer for catching this error. Correction has been made in Figure 3.

Figure R15. Demonstration of different modes of iCarP illumination on rat hearts. a,b, iCarP supporting color change in illumination, the same device delivered light of (i) 405 nm, (ii) 520 nm, (iii) 660 nm. c,d, Pulsatile illumination at 1 Hz, (i), (ii), (iii) shows the on-off-on switch. e,f, iCarP illumination with increasing intensity over time. White arrows indicate the optical fibers. **Scale bar = 200 μ m.** (from Fig. 3 in Manuscript)

References

1. Fang, W., Yang, L., Hong, L. & Hu, Q. A chitosan hydrogel sealant with self-contractile characteristic: From rapid and long-term hemorrhage control to wound closure and repair. *Carbohydr. Polym.* **271**, 118428 (2021).
2. Li, W., Zhong, D., Hua, S., Du, Z. & Zhou, M. Biomaterialized Biohybrid Algae for Tumor Hypoxia Modulation and Cascade Radio-Photodynamic Therapy. *ACS Appl. Mater. Interfaces* **12**, 44541–44553 (2020).
3. Qiao, Y. *et al.* Engineered algae: A novel oxygen-generating system for effective treatment of hypoxic cancer. *Sci. Adv.* **6**, eaba5996 (2020).
4. Yamagishi, K. *et al.* Tissue-adhesive wirelessly powered optoelectronic device for metronomic photodynamic cancer therapy. *Nat. Biomed. Eng.* **3**, 27–36 (2018).
5. Lee, G.-H. *et al.* Multifunctional materials for implantable and wearable photonic healthcare devices. *Nat. Rev. Mater.* **5**, 149–165 (2020).
6. Nussinovitch, U. & Gepstein, L. Optogenetics for in vivo cardiac pacing and resynchronization therapies. *Nat. Biotechnol.* **33**, 750–754 (2015).
7. Prabhu, S. D. & Frangogiannis, N. G. The Biological Basis for Cardiac Repair After Myocardial Infarction: From Inflammation to Fibrosis. *Circ. Res.* **119**, 91–112 (2016).
8. Wu, Q. *et al.* Extracellular vesicles from human embryonic stem cell-derived cardiovascular progenitor cells promote cardiac infarct healing through reducing cardiomyocyte death and promoting angiogenesis. *Cell Death Dis.* **11**, 354 (2020).
9. Lin, X. *et al.* A viscoelastic adhesive epicardial patch for treating myocardial infarction. *Nat. Biomed. Eng.* **3**, 632–643 (2019).
10. Montgomery, M. *et al.* Flexible shape-memory scaffold for minimally invasive delivery of functional tissues. *Nat. Mater.* **16**, 1038–1046 (2017).
11. Wei, X. *et al.* Hybrid Integrated Cardiomyocyte Biosensors for Bitter Detection and Cardiotoxicity Assessment. *ACS Sens.* **6**, 2593–2604 (2021).

REVIEWER COMMENTS

Reviewer #1 (Remarks to the Author):

The current status of this manuscript is suitable for publishing on Nature Communications.

Reviewer #2 (Remarks to the Author):

The revised version of this paper has added new results, and although this demonstrates an upgrade in fiber optic technology, it is neither deep nor innovative enough in terms of molecular mechanics, which may not reach the level of this journal.

1. The representative images of cardiomyocyte apoptosis in Figure R9 are problematic. Firstly, these fluorescent images are blurred and of low quality. Secondly, Bcl-2 in the MI group and Bax and Caspase-3 in the Sham group showed no fluorescence signal at all, which was unrealistic and untrustworthy. Finally, the histogram lacks statistical data of the Sham group, and the grouping of representative image and statistical chart need to be consistent.

2. As described in the article, iCarP is used in tumor therapy at higher temperatures and larger illumination area, but photothermal treatment can kill the tumor, while also "injuring" the normal tissue near the tumor. How can iCarP perform precise treatment without damaging the surrounding tissues? In this study, we do not seem to see the advantages of iCarP in this regard.

3. In Figure 5C, it is difficult to see the difference in cardiomyocyte apoptosis among the groups by TUNEL histochemical staining, and it is recommended to replace them with more representative.

4. In Figure R4. The authors evaluated only 5 minutes ECG is not representative, this time is very short and cannot really evaluate the effect of iCarP on arrhythmia in animals. This study lacks a systematic and long-term analysis of iCarP on arrhythmias in animals, such as the detection of arrhythmia indicators (PR, QRS, QTC, etc.) after MI surgery, and the detection of arrhythmia susceptibility by programmed electrical stimulation.

5. The description of the section "iCarP supported photosynthesis for myocardial treatment" is confusing. The role of Chlorella and its research background are not clearly explained in the manuscript, leaving readers confused about its role in the treatment of myocardial infarction by iCarP.

6. According to the results of RNA-seq in Figure R7e, the expression level of Bax in iCarP+/light+ group did not change significantly compared with the MI group, which seems to be contradictory with the results in Figure R9b. Please explain this problem.

7. As shown in Figure R12, it is difficult to effectively and adequately assess the damage to the heart caused by prolonged light exposure only by detecting cell viability at the in vitro level. It is better to conduct comprehensive confirmation in vivo.

Reviewer #3 (Remarks to the Author):

The authors responded well and addressed the questions raised, suggesting acceptance.

Response to reviewers

We greatly appreciate the reviewer for additional questions, particularly the ones regarding the safety of iCarP illumination. We hope our responses listed below have answered the questions well.

1. The representative images of cardiomyocyte apoptosis in Figure R9 are problematic. Firstly, these fluorescent images are blurred and of low quality. Secondly, Bcl-2 in the MI group and Bax and Caspase-3 in the Sham group showed no fluorescence signal at all, which was unrealistic and untrustworthy. Finally, the histogram lacks statistical data of the Sham group, and the grouping of representative image and statistical chart need to be consistent.

Answer: We thank the reviewer for raising this question. The fluorescence signals of Bcl-2 in the MI group and Bax and Caspase-3 in the Sham group were not obvious to naked eye due to the low brightness of the original images, despite that they could be recognized by ImageJ. We apologize for the confusion and have increased the brightness of all fluorescent images in **Supplementary Fig. 9** by the same percentage. The green fluorescence signal of Bcl-2 in the MI group and Bax and Caspase-3 in the Sham group can be clearly seen as shown in **Fig. R1**. Enlarged figures were presented in **Fig. R1g**. In addition, statistical data of the Sham group was added to the histogram.

Figure in the supporting information (**Supplementary Fig. 9**) was updated accordingly, at page 18 line 302.

Figure R1. Expression of apoptosis markers in myocardium. Representative immunofluorescence staining images of cTnT /Bcl-2 (a), cTnT /Bax (b) and cTnT /Caspase-3 (c) in left ventricular 1 d after MI (nuclei: blue, cTnT: red, Bcl-2/Bax/Caspase-3: green). Scale bars = 200 μ m. Quantitative analysis of expression of Bcl-2 (d), Bax (e) and Caspase-3 (f). **g**, Large vision of Bcl-2 in the MI group and Bax and Caspase-3 in the Sham group. Scale bars = 200 μ m.

2. As described in the article, iCarP is used in tumor therapy at higher temperatures and larger illumination area, but photothermal treatment can kill the tumor, while also “injuring” the normal tissue near the tumor. How can iCarP perform precise treatment without damaging the surrounding tissues? In this study, we do not seem to see the advantages of iCarP in this regard.

Answer: We thank the reviewer for raising this question. In **Fig. 3**, we showed the advantage of iCarP in achieving large illumination areas and illumination depths compared to traditional optical fiber/patch waveguides and clinically used optical fibers. iCarP scatters optical energy in the forward direction of the optical fiber to lateral directions, and the illumination scope of iCarP could be adjusted by changing the size of PMCL patch. This means that by controlling the diameter of PMCL substrate, one can match the size of the illumination area with the size of the target tissue, and minimize the damage to adjacent normal tissue.

Controlling iCarP illumination area by PMCL diameter is shown in **Fig. R2** from added experiment. A large area (diameter >2.5 cm) of superficial myocardium of an ex vivo porcine heart was injected with Au nanorods, followed by adhesion of iCarP with different diameters. The heart was illuminated with 808 nm light and observed by thermal imager. As expected, the majority of the heated area was directly beneath iCarP, and only a small portion was on the edge of iCarP due to light scattering in the tissue (light did not increase the temperature of free-standing iCarP, data not shown). The round shape of heated myocardium matched with the round patches. The diameter of the heated tissue increased with iCarP diameter, which demonstrated that one can match the illumination area with the target size by controlling iCarP diameter, thus minimizing undesired illumination.

Figure R2. Infrared thermography and digital images (insets) of an ex vivo porcine heart illuminated and heated by iCarPs with diameters of 0.5 (a), 1.5 (b) and 2.5 (c) cm. Scale bars: white = 1 cm, black = 2 cm.

Text in the Results section of the manuscript was modified accordingly, at page 19 line 516.

“When vertically inserted into the porcine myocardium, the emitted light illuminated a column volume with a ~2 cm diameter and depth equal to 0.5 cm plus fiber insertion length. One can control iCarP illumination area by adjusting patch diameter, to achieve precise treatment and minimize influence on adjacent tissue.”

3. In Figure 5C, it is difficult to see the difference in cardiomyocyte apoptosis among the groups by TUNEL histochemical staining, and it is recommended to replace them with more representative.

Answer: We appreciate the reviewer for the suggestion. We have updated the TUNEL staining images, with larger magnification insets, to better show the differences. Figure and caption in **Fig. 5c** of the figures was modified accordingly.

Figure R3. Representative TUNEL staining images of heart sections 3 d after MI. Scale bars: black = 5 mm, red = 500 μm , blue = 50 μm .

4. In Figure R4. The authors evaluated only 5 minutes ECG is not representative, this time is very short and cannot really evaluate the effect of iCarP on arrhythmia in animals. This study lacks a systematic and long-term analysis of iCarP on arrhythmias in animals, such as the detection of arrhythmia indicators (PR, QRS, QTC, etc.) after MI surgery, and the detection of arrhythmia susceptibility by programmed electrical stimulation.

Answer: We agree with the reviewer that evaluating the risk of iCarP illumination causing arrhythmia is important. Safety of cardiac illumination or cardiac phototherapies is a question for the entire research field.

In addition to the canine experiment included in the last revision (with which we showed that no arrhythmia was observed within 5 min post illumination, on both non-infarcted and infarcted hearts), we recorded and analyzed the ECG signals for 3 days after iCarP illumination on rat hearts. As shown in **Fig. R4**, at 0h, 9h, 1d and 3d after illumination, PR, QT, QRS and QTc of the Sham group did not significantly change from the baseline (indicators recorded in healthy rats). In addition, the indicators at later time points did not vary from the indicators at early time points, showing that iCarP illumination did not increase long-term risk of arrhythmia. In MI group, the indicators QT, QRS and QTc increased from the earliest time point post illumination, and such increase lasted in the entire observation period, and did not enlarge. On the other hand, PR increased at early time points and returned to baseline. Therefore, it could be concluded that the changes in indicators of MI group is attributed to ischemia in the myocardium, not illumination. In terms of detection of arrhythmia susceptibility by programmed electrical stimulation, it is currently not among our expertise. We will establish the capability of programmed electrical stimulation and answer this part of the question in future studies. We searched the literature, among recently published 150 research papers about using conductive biomaterials to improve electrical signal conduction in infarcted myocardium, only 6 (4%) evaluated the effect of biomaterials on myocardial susceptibility by programmed electrical stimulation. It seems that ECG only is widely considered enough to evaluate the risk of arrhythmia or influence of different treatments on electrical signal conduction.

Combined with the canine short-term study (results from last revision copied below the rat data), the rat long-term experiments showed that the arrhythmia risk caused by illumination is low. From other cardiac illumination studies, we found support for this conclusion. Nussinovitch and Gepstein used adeno-associated virus (AAV) 9 to express the Channelrhodopsin-2 (ChR2) transgene at one or more ventricular sites in rats, so that to use optogenetics for in vivo cardiac pacing¹ (Nussinovitch et al, *Nat. Biotechnol.* 2015). The authors found that only the myocardium expressing ChR2 was directly responding to illumination, illumination on areas not expressing ChR2 did not change the heart rate. No arrhythmia by illumination on areas not expressing ChR2 was reported. Similarly, recent work by Deisseroth et al. achieved cardiogenic control of affective behavioural state using optogenetics² (Deisseroth et al, *Nature*, 2023). High frequency flashing was used to increase heart rate and mimic arrhythmia, no arrhythmia after high frequency flashing was reported. Woo et al. used *S. elongatus* as the photosynthesis unit to provide oxygen to ischemic myocardium. No arrhythmia or other adverse responses were reported after direct illumination by a light bulb during the open chest surgery³ (Woo et al, *Sci. Adv.* 2017).

Different from the studies mentioned above, our study is mainly focused on developing a photonic device to achieve large area, high intensity, wide spectrum, continuous/pulsatile, deeply penetrating illumination. The tumor therapies and MI treatment demonstrations are to show the potential of iCarP in in vivo applications. For example, if one wants to translate Prof. Deisseroth's recent finding (Deisseroth et al, *Nature*, 2023) from mice to clinical applications², iCarP could be employed for patient heart illumination. Although extensive study of physiological and biological influences is not the focus of our study, we believe our results provided new evidence for safety of cardiac illumination.

Figure R4. Electrocardiograph analysis. **a**, Representative ECGs of rats in MI and Sham group at different times after iCarP illumination. PR interval duration (**b**), QRS interval duration (**c**), QT interval duration (**d**) and QTc interval duration (**e**) of rats in MI and Sham group at different times after iCarP illumination.

Figure R5. Effects of iCarP on heart rate of a dog. **a**, Preoperative preparation. **b**, Thoracotomy & iCarP implantation. **c**, Ligation of LAD and LCx. **d**, iCarP implantation & illumination. **e-l**, Dog heart rate measured by electrocardiogram (**e-h**) and U-iCarP (**i-l**) at different stages of the surgery, including before illumination (before ligation) (**e,i**), during illumination (before ligation) (**f,j**), before illumination (after ligation) (**g,k**) and during illumination (after ligation) (**h,l**). **m**, Structural diagram of U-iCarP. **n**, Digital image of U-iCarP. **o**, Application of U-iCarP on the dog heart.

Text in the Results section of the manuscript was modified accordingly, at page 24 line 665.

“The influence of iCarP illumination on heart rate of healthy and MI dogs and rats was evaluated. As shown in Supplementary Fig. 14 and Supplementary Video 7, iCarP illumination did not affect heart rate of the dog, or cause arrhythmia. Consistently, iCarP illumination did not affect heart rate of rats, or cause arrhythmia within 3 days of observation (data not shown). These results showed the adaptability and safety of iCarP with minimally invasive implantation surgeries on other internal organs and tissues.”

5. The description of the section "iCarP supported photosynthesis for myocardial treatment" is confusing. The role of Chlorella and its research background are not clearly explained in the manuscript, leaving readers confused about its role in the treatment of myocardial infarction by iCarP.

Answer: We appreciate the reviewer for the suggestion to enrich the Chlorella background.

In myocardial infarction, the occlusion of coronary artery causes acute ischemia of the myocardium, leading to the unbalanced consumption of oxygen and adenosine triphosphate, and ischemic necrosis of the myocardium⁴ (Heusch, *Am. J. Physiol.-Heart Circ. Physiol.* 2019). Biocompatible biomaterials including hemoglobin-based oxygen carriers, perfluorocarbons, peroxides have been applied to provide oxygen to ischemic myocardium. Taking into account both oxygen production and consumption of metabolites, photosynthesis units reuse the metabolic products and produce O₂⁵⁻⁷ (Khavari et al, *Mol. Biol. Rep.* 2021; Chen et al, *Chem. Soc. Rev.* 2021; Wang et al. *Bio-Des. Manuf.* 2021). For example, Haraguchi et al. created an in vitro “symbiotic recycling system” composed of mammalian cells and algae, which can increase the dissolved oxygen concentration in the medium and decrease glucose consumption and lactate production, resulting in thicker cardiac tissues⁸ (Haraguchi et al, *Sci. Rep.* 2017). Woo et al. directly applied photosynthetic *S. elongatus* to infarcted myocardium and introduced in situ photosynthesis³ (Woo et al, *Sci. Adv.* 2017). *S. elongatus* under local light irradiation provided oxygen and nutrients to ischemic myocardium, thus alleviated the ischemic conditions. Zhang et al. designed highly efficient a opto-driven hybrid proteoliposome system by combining thylakoid fragments with lipid molecules, which can rescue mouse heart from myocardial infarction⁹ (Zheng et al, *Adv. Mater.* 2018). Chlorella as a unicellular microalga contains a high concentration of chlorophyll, which absorbs light across a broad wavelength spectrum for photosynthesis. Combined with its good biocompatibility, Chlorella showed a good potential in treating progressive diseases^{10,11} (Qiao et al, *Sci. Adv.* 2020; Zhou et al, *ACS Appl. Mater. Interfaces* 2020).

Text in the Results section of the manuscript was modified accordingly, at page 21 line 566.

“Such theoretical advantage of iCarP was evaluated in in situ photosynthesis for MI treatment (Fig. 5a). In previous studies, safety and efficacy of algae-based photosynthesis systems have been demonstrated in treating hypoxia diseases^{47,48}. The entire infarcted rat myocardium turned from pale white to light green after intramyocardial Chlorella injection, showing that the injected Chlorella distributed throughout the infarct (data not shown).”

6. According to the results of RNA-seq in Figure R7e, the expression level of Bax in iCarP+/light+ group did not change significantly compared with the MI group, which seems to be contradictory with the results in Figure R9b. Please explain this problem.

Answer: We thank the reviewer for raising this question. **Fig. R7e** (in the last revision) is to evaluate Bax RNA level, **Fig. R9b** (in the last revision) is to evaluate BAX protein translation level. They showed the same trend (MI group higher than iCarP+/Light+ group). The difference is that in **Fig. R7e**, RNA-seq did not detect significant difference between the two groups, while in **Fig. R9b**, immunofluorescent staining detected significant difference between the two groups.

Recent research indicates that cellular protein levels are not necessarily correlated with the RNA levels¹²⁻¹⁵ (Khan et al, *Science* 2013; Wu et al, *Nature* 2013; He et al, *Nat. Rev. Genet.* 2014; Klapproth et al, *Nat. Commun.* 2022), which emphasizes the importance of post-transcriptional regulation of gene expression. It is not rare that comparison of RNA levels and protein levels of the same gene among different groups show the same trend but do not match in evaluating the significance of differences. In many cases, the RNA level comparison and protein level comparison even show opposite trends.

Protein translation efficiency is not only regulated by RNA concentration. Protein production is also subjected to various types of post-transcriptional regulation — such as through mRNA secondary structure, microRNAs or mRNA translational control — which probably contributes as much as or even more than, transcriptional regulation to determine cellular protein abundance¹⁴ (He et al, *Nat. Rev. Genet.* 2014). For example, a lot of recent studies have shown that RNA modifications including RNA methylation post-transcriptionally regulate protein expression via affecting RNA stability, localization, transport, splicing and translation.

In addition to post-transcriptional regulation, the difference in the significance levels between RNA level comparison and protein level comparison (MI group vs iCarP+/Light+ group) might also be a result of the systemic errors of the RNA-seq and immunofluorescent staining techniques.

Text in the Results section of the manuscript was modified accordingly, at page 22 line 597.

“...RNA-seq showed that the expression levels of representative pro-apoptosis genes in LV myocardium, including Bax, Htra2 and Myc were elevated by MI compared to baseline levels in Sham rats, and were decreased by in situ photosynthesis in iCarP+/Light+ group (although showing the trend, significant difference in BAX level between MI and iCarP+/Light+ groups was not detected, which is probably due to the difference in post-transcriptional regulation or systemic errors of the RNA-seq and immunofluorescent staining techniques). In vitro cell culture showed that illumination (660 nm, 55 mW, 3 h) with or without Chlorella did not induce intracellular ROS accumulation in cardiomyocytes...”

7. As shown in Figure R12, it is difficult to effectively and adequately assess the damage to the heart caused by prolonged light exposure only by detecting cell viability at the in vitro level. It is better to conduct comprehensive confirmation in vivo.

Answer: We thank the reviewer for raising this question. Considering required duration for most phototherapies, and the risk of animal death after prolonged anesthesia, we think 6 h illumination in vivo is enough to evaluate damage to the heart caused by prolonged light exposure. After adhesion of iCarP using CCS@Gel, continuous illumination (660 nm, 55 mW) was performed for 6h. One day later, the rats were euthanized and rat hearts were fixed with 4% paraformaldehyde, embedded in paraffin, and sectioned followed by TUNEL staining to assess cell apoptosis. The results shown in **Fig. R6** demonstrated that 6 h continuous illumination did not cause significant damage to the heart. Combined with evidence provided in the original manuscript and first revision and answer to Question 5 regarding the risk of arrhythmia post illumination, it could be concluded that iCarP illumination (660 nm, 55 mW) is safe to rat hearts.

As mentioned in the answer to Question 5, studies have illuminated hearts with different methods. Nussinovitch and Gepstein illuminated rat hearts by a 2 mm LED-coupled optic fiber in open or closed chest surgeries, prolonged illumination lasted 2 h¹ (Nussinovitch et al, *Nat. Biotechnol.* 2015). Deisseroth et al. illuminated mice heart with a vest containing a micro-LED, which was fastened to the animals' chests over the hearts² (Deisseroth et al, *Nature* 2023, **Fig. R6f**). In these studies, damages to animal hearts by illumination were not reported.

In terms of safety of cardiac illumination, we believe that our study provided new evidence from in vivo works to the research field, including characterization of cardiomyocyte deaths after 6 h prolonged illumination, 3-day follow-up of arrhythmia indicators post illumination, evaluation of risks of inducing arrhythmia in infarcted canine hearts.

Figure R6. Representative TUNEL staining images in rat hearts 1 d after 6 h iCarP illumination. **a**, Panoramic image, scale bar = 2 mm. **b**, Enlarged image, scale bars = 200 μ m. **c**, Enlarged image, scale bars = 20 μ m. **d**, Quantitative analysis of apoptotic cells based on TUNEL staining, scale bar = 20 μ m. **e**, Optogenetic pacing in an open-chest model (i) and a closed-chest model (ii), adapted from *Nat. Biotechnol.* 2015; 33: 750–754¹. **f**, Controlling anxiety with a non-invasive optical pacemaker, adapted from *Nature* 2023; 615: 292–299².

Text in the Results section of the manuscript was modified accordingly, at page 22 line 603.

“In vitro cell culture showed that illumination (660 nm, 55 mW, 3 h) with or without Chlorella did not induce intracellular ROS accumulation in cardiomyocytes, as shown in Supplementary Fig. 11. Consecutive illumination for 12 h did not decrease the viability of cardiomyocytes in vitro, and 6 h consecutive illumination did not cause damage to rat hearts in vivo, indicating the safety of extended iCarP illumination.”

References

1. Nussinovitch, U. & Gepstein, L. Optogenetics for in vivo cardiac pacing and resynchronization therapies. *Nat. Biotechnol.* **33**, 750–754 (2015).
2. Hsueh, B. *et al.* Cardiogenic control of affective behavioural state. *Nature* **615**, 292–299 (2023).
3. Cohen, J. E. *et al.* An innovative biologic system for photon-powered myocardium in the ischemic heart. *Sci. Adv.* **3**, e1603078 (2017).
4. Heusch, G. Myocardial ischemia: lack of coronary blood flow, myocardial oxygen supply-demand imbalance, or what? *Am. J. Physiol.-Heart Circ. Physiol.* **316**, H1439–H1446 (2019).
5. Khavari, F., Saidijam, M., Taheri, M. & Nouri, F. Microalgae: therapeutic potentials and applications. *Mol. Biol. Rep.* **48**, 4757–4765 (2021).
6. Chen, Q.W., Qiao, J.Y., Liu, X.H., Zhang, C. & Zhang, X.Z. Customized materials-assisted microorganisms in tumor therapeutics. *Chem. Soc. Rev.* **50**, 12576–12615 (2021).
7. Wang, Y. *et al.* Photosynthetic biomaterials: applications of photosynthesis in algae as oxygenerator in biomedical therapies. *Bio-Des. Manuf.* **4**, 596–611 (2021).
8. Haraguchi, Y. *et al.* Thicker three-dimensional tissue from a “symbiotic recycling system” combining mammalian cells and algae. *Sci. Rep.* **7**, 41594 (2017).
9. Zheng, D.W. *et al.* Photo-Powered Artificial Organelles for ATP Generation and Life-Sustainment. *Adv. Mater.* **30**, 1805038 (2018).
10. Qiao, Y. *et al.* Engineered algae: A novel oxygen-generating system for effective treatment of hypoxic cancer. *Sci. Adv.* **6**, eaba5996 (2020).
11. Li, W., Zhong, D., Hua, S., Du, Z. & Zhou, M. Biomaterialized Biohybrid Algae for Tumor Hypoxia Modulation and Cascade Radio-Photodynamic Therapy. *ACS Appl. Mater. Interfaces* **12**, 44541–44553 (2020).
12. Khan, Z. *et al.* Primate Transcript and Protein Expression Levels Evolve Under Compensatory Selection Pressures. *Science* **342**, 1100–1104 (2013).
13. Wu, L. *et al.* Variation and genetic control of protein abundance in humans. *Nature* **499**, 79–82 (2013).
14. Fu, Y., Dominissini, D., Rechavi, G. & He, C. Gene expression regulation mediated through reversible m6A RNA methylation. *Nat. Rev. Genet.* **15**, 293–306 (2014).
15. Klapproth, E. *et al.* Targeting cardiomyocyte ADAM10 ectodomain shedding promotes survival early after myocardial infarction. *Nat. Commun.* **13**, 7648 (2022).

REVIEWER COMMENTS

Reviewer #2 (Remarks to the Author):

The revised version of this paper has been improved to some extent. For the benefit of the reader, however, there are still some minor issues that need attention to modify. There are given below.

1. In Figure R1, please specify whether this Caspase-3 is the total or Cleaved-Caspase 3 form. If this is an immunofluorescence result for total Caspase 3, it is recommended to supplement Western-Blot experiments to clarify the change in Cleaved-Caspase 3 expression
2. In Figure 4I (i-iv), the nuclei are barely visible in the middle three groups of TUNEL and HE staining results. If the representative diagram is selected incorrectly, please explain or choose a more representative diagram.
3. In Figure5C TUNEL results, the number of nuclei in the Sham group was significantly less than that in the other three groups. Was it observed under the same magnification?
4. In Figure5D, the histogram lacks statistical data of the Sham group, and the grouping of representative image and statistical chart need to be consistent.

Response to reviewers

We greatly appreciate reviewer 2's comments and his/her assistance in improving our manuscript. We carefully went through the 4 questions reviewer raised in the third round, and organized our answers in the following pages.

Question 1. In Figure R1, please specify whether this Caspase-3 is the total or Cleaved-Caspase 3 form. If this is an immunofluorescence result for total Caspase 3, it is recommended to supplement Western-Blot experiments to clarify the change in Cleaved-Caspase 3 expression.

Answer: We thank the reviewer's question regarding the type of antibody used to stain Caspase-3. We described in SI that the antibody used is for cleaved Caspase-3, purchased from CST, US (SI page 6, line 139).

135 **Supplementary Note 9. Immunofluorescent staining**

136 Rats were anesthetized and sacrificed at 1 d post MI to harvest treated hearts. Rat hearts were
137 fixed with 4% paraformaldehyde, embedded in paraffin, and sectioned. The hearts were
138 immunofluorescently stained with DAPI, cTnT and Bcl-2 (Abcepta, China) /Bax (Abcam, UK)
139 /Cleaved Caspase-3 (CST, US) to reveal apoptosis. Images were captured using DP2-BSW software
140 (Olympus, VS200, USA), and analyzed by ImageJ software.

We added “cleaved” to related text, figures and captions, the modified contents are pasted below.

Text in the Materials and methods section of the manuscript was modified accordingly, at page 6 line 136, and page 22 line 599.

“Dulbecco's modified Eagle's medium (DMEM) (Gibco, USA), fetal bovine serum (Sijiqing, China), Cell counting kit-8 (CCK-8) and dichloro-dihydro-fluorescein diacetate (DCFH-DA) (Beyotime Biotechnology, China), One Step TUNEL Apoptosis Assay Kit (Beyotime Biotechnology, China), Anti-Connexin 43/GJA1 antibody-Intercellular Junction Marker (Acabam, UK), Anti-Cardiac Troponin T antibody (Acabam, UK), Anti-Bcl-2 Antibody (Abcepta, China), Anti-Bax

Antibody (Abcam, UK) and Cleaved Caspase-3 Rabbit mAb (CST, USA) were used according to corresponding protocols.”

“Immunofluorescent staining of apoptosis related molecules Bcl-2, Bax, and Cleaved Caspase-3 exhibited consistent results. As shown in Supplementary Fig. 9, expression level of anti-apoptotic Bcl-2 was reduced by MI compared to that in Sham group, and was highest in iCarP+/Light+ group. In addition, the expression levels of pro-apoptotic Bax and Cleaved Caspase-3 were lowest in iCarP+/Light+ group,”

Supplementary Figure 9 in Supporting information was modified accordingly

Supplementary Figure 9. Expression of apoptosis markers in myocardium. Representative immunofluorescent staining images of cTnT/Bcl-2 (**a**), cTnT/Bax (**b**) and cTnT/Cleaved Caspase-3 (**c**) in left ventricular 1 d after MI (nuclei: blue, cTnT: red, Bcl-2/Bax/Cleaved Caspase-3: green). Scale bars = 200 μ m. Quantitative analysis of expression of Bcl-2 (**d**), Bax (**e**) and Cleaved Caspase-3 (**f**). Statistical significance was calculated using ANOVA and data are presented as means \pm SEM. **p < 0.01, ****p < 0.0001.

Question 2. In Figure 4I (i-iv), the nuclei are barely visible in the middle three groups of TUNEL and HE staining results. If the representative diagram is selected incorrectly, please explain or choose a more representative diagram.

Answer: Less nuclei was found in the photothermally treated area of tumor because a high percentage of cells were dead and failed to maintain the integrity of their nuclei. This phenomenon is extensively reported in literature (please see the figures from the literatures below).

Advanced Materials, 2022, 34, 2108146

Advanced Materials, 2022, 34, 2107009.

Figure 4I from our manuscript is pasted below:

We added the explanation in the Results section, at page 20 line 545.

“Both TUNEL and H&E staining showed that severe cell apoptosis occurred in the center 2 mm (long axis) hemi-ellipsoid volume of the tumor tissue 2 days after a single 5 min iCarP illumination (Fig. 4I). Less nuclei were found in the photothermally treated area because the cells were dead and failed to maintain the integrity of their nuclei. In comparison, the necrosis region was a 1 mm diameter column in tumor 2 days after a single illumination by flat-end optical fibers, smaller than the hemi-ellipsoids in iCarP treated tumors (Fig. 4m).”

Question 3. In Figure 5C TUNEL results, the number of nuclei in the Sham group was significantly less than that in the other three groups. Was it observed under the same magnification?

Answer: The images were recoded under the same magnification. The number of nuclei in the Sham group was significantly smaller because in the MI and two treatment groups, there are high density inflammatory cells as a result of the ischemic injury. Whereas in Sham group, there were few inflammatory cells as there was no ischemic injury, only healthy cardiomyocytes whose volume is bigger compared to inflammatory cells. The cell nuclei densities in the remote areas (relatively healthy areas) were similar to the Sham group, please find the images below, in which we zoomed in the remote areas of the treatment groups (same magnification as the Sham group).

Healthy areas: mainly cardiomyocytes, bigger cell volume, low cell density

Injured areas: mainly inflammatory cells, smaller cell volume, high cell density

Text in the Results section of the manuscript was modified accordingly, at page 22 line 596.

“Between the two iCarP treated groups, the iCarP+/Light+ group showed a lower percentage of apoptotic cells compared to iCarP+/Light- group (Fig. 5d), attributed to oxygen produced by light triggered photosynthesis. The cell densities in the infarcted myocardium of MI and two treatment groups were higher compared to Sham myocardium, which was due to recruitment of

inflammatory cells in the infarcts. Immunofluorescent staining of apoptosis related molecules Bcl-2, Bax, and Cleaved Caspase-3 exhibited consistent results.”

Yoshimuzi et al. followed the histological changes after MI injury in rat hearts. High density of inflammatory cells can be seen in the infarct at 1 w and 2 w post MI (*Biomaterials*, 2016, 83, 182-193), the figure is attached below.

Question 4. In Figure5D, the histogram lacks statistical data of the Sham group, and the grouping of representative image and statistical chart need to be consistent.

Answer: We added Sham group to the histogram, as attached below. The updated figure and text are pasted below.

Text in the Results section of the manuscript was modified to describe results of the Sham group, at page 21 line 590, page 22 line 615, page 23 line 628.

“Cardiac cell apoptosis 3 days post-MI was detected by TUNEL staining (Fig. 5c). The ischemic injury caused significant cell death in the MI group compared to Sham control. The percentage of apoptotic cardiac cells significantly decreased in iCarP+/Light- group compared to the MI group, showing that iCarP alone can inhibit cardiac cell death by providing mechanical support (Fig. 5d).”

“Myocardial fibrosis detected by Masson’s trichrome staining 28 days post-surgery (Fig. 5e) exhibited a trend same to apoptosis results, which is expected as fibrosis is considered a result of cardiac damage⁴⁹. Compared to MI group (25.6±2.8%), iCarP+/Light- treatment significantly reduced cardiac fibrosis (22.1±2.0%), while iCarP+/Light+ group had the lowest fibrosis level (20.1±2.5%).”

“Left ventricle ejection fraction (LVEF) significantly decreased in MI group compared to Sham, indicating cardiac dysfunction after myocardial infarction. Although there was an increasing trend, LVEF of iCarP+/Light- group (with theoretical mechanical support) was not significantly higher compared to LVEF of MI group, despite the short-term cell protection effect shown in TUNEL staining.”

REVIEWERS' COMMENTS

Reviewer #2 (Remarks to the Author):

The authors responded well and addressed the issues raised, suggesting acceptance.